# Cold Start in the Dark: Efficient and Practical Model Extraction of GNNs

## Abstract

The deployment of Graph Neural Networks (GNNs) on MLaaS platforms makes them vulnerable to Model Extraction Attacks (MEAs), where an adversary queries a proprietary model's API to reconstruct a high-fidelity surrogate. However, the practicality of current methods is limited by unrealistic assumptions, such as access to detailed soft-label probabilities, large initial seed datasets, or permissive query budgets. To address this gap, this work introduces MIME, a framework designed for the more stringent and realistic "Cold Start in the Dark" problem, where an adversary operates with no initial labels and only hard-label feedback under a tight budget. MIME resolves the critical cold start challenge using unsupervised pre-training to establish a strong structural baseline from the topology alone. This bootstraps a query-efficient active learning loop that strategically balances node uncertainty and diversity, ensuring robustness through adaptive graph regularization. Extensive experiments show that MIME achieves strong performance on both model accuracy and fidelity. The findings demonstrate a practical and stealthy attack vector, exposing a concrete security risk to production GNNs by succeeding under realistic adversarial constraints.

## 1 Introduction

Graph Neural Networks (GNNs) are foundational to modern machine learning, delivering state-of-the-art results on graph-structured data in fields like social network analysis (Gupta et al., 2021), molecular chemistry (Scarselli et al., 2009), and computational biology (Muzio et al., 2021). Their success has fueled the rise of Graph-based Machine-Learning-as-a-Service (GMLaaS) platforms that offer access to proprietary GNNs via pay-per-query APIs (Wu et al., 2024). These models, representing significant investments in data, computation, and expertise, become vulnerable to theft when exposed through public APIs (Dubey et al., 2022; Hou et al., 2019).

The MLaaS platforms present a core conflict between profitability and security (Kesarwani et al., 2018). While pay-per-query interfaces are essential for revenue, they also create a primary attack vector (Carlini et al., 2021). Model Extraction Attacks (MEAs) are a major threat in this context, driven by clear economic incentives: if the cost of replicating a model is less than the cost of using the service, an adversary is motivated to act (Dubey et al., 2022; Gong et al., 2020). Attackers can use black-box queries to train a substitute model, effectively stealing the provider's intellectual property (Tramer et al., 2016; Orekondy et al., 2019; Wang & Gong, 2018). A successful MEA is not just a loss of IP but a security breach, as the stolen model can be used for further attacks, such as crafting adversarial inputs or inferring sensitive training data (Chandrasekaran et al., 2020).

Despite the clear threat posed by MEAs, a significant gap persists between their theoretical potential and practical execution, particularly for Graph Neural Networks (GNNs). This gap stems from a research paradigm built on unrealistic assumptions, revealing four foundational flaws when the idealized conditions of the literature are contrasted with the harsh constraints of a real-world attack.

**Data Assumption.** Many model extraction attacks assume a data-rich adversary with a large, unlabeled dataset similar to the victim's (Orekondy et al., 2019; He et al., 2023). A more realistic setting, however, is data scarcity, where an attacker has only a small, random sample of nodes. Such a sample is insufficient to infer the data distribution and lacks strategic value.

**Feedback Assumption.** Much of the literature assumes attackers receive soft-label feedback—informative probability vectors that show model confidence and offer a strong learning signal (Orekondy et al., 2019; Krishna et al., 2020). In reality, providers typically return only hard labels, giving just the top-1 prediction. This minimal, non-differentiable feedback significantly increases the difficulty of an attack.

**Query Assumption.** Existing methods often assume a permissive query budget, allowing for a large number of queries without restriction (He et al., 2021). Real-world scenarios are far more constrained: APIs use rate limiting and anomaly detection (Cheng et al., 2025), and the pay-per-query model imposes a strict financial budget. This forces an attacker to be highly efficient, often limited to single-node queries (Wang et al., 2022).

**Seed Assumption.** Many attacks require an initial set of seed nodes with known ground-truth labels to begin (Kipf & Welling, 2017; Zhang et al., 2022). In a true black-box setting, an adversary has no such labels. Their only source of supervision comes from the victim model's own predictions, which serve as potentially flawed pseudo-labels, offering no initially trusted information.

These assumptions are not isolated; they are interconnected "crutches" that have enabled prior work to bypass significant challenges (Wu et al., 2023). A large dataset reduces the need for a sophisticated query strategy, soft labels facilitate a seed-free start, and an unlimited query budget allows for brute-force solutions. When these supports are removed, the attacker faces a dilemma: an effective query strategy is needed, but developing one is nearly impossible with a tight budget, hard labels, and no trusted seeds for guidance (Li et al., 2018). We formalize this challenge as the **Cold Start in the Dark** problem, describing an adversary who starts "cold" (with no ground-truth labels) and operates "in the dark" (with scarce data, hard-label feedback, and a strict query budget). Conventional strategies fail against this realistic benchmark, necessitating a new approach.

To address the **Cold Start in the Dark** problem, we introduce **MIME** (**Minimal Information Model Extraction**), a framework engineered to operate with minimal information: no proxy data, no initial labels, hard-label feedback, and a tight query budget. MIME strategically builds knowledge from scratch. First, it uses unsupervised pre-training (Deep Graph Infomax) to learn from the graph's structure alone, then efficiently queries nodes by sequentially filtering for uncertainty and diversity. Subsequently, a surrogate model is trained on the acquired hard labels, stabilized by a composite loss that leverages graph topology to compensate for sparse data. Finally, post-budget self-training refines the model at no additional cost.

**Contributions.** Our key contributions are as follows. (i) Problem formulation: We define the Cold Start in the Dark problem, a stringent and practical attack scenario for GNNs that reflects real-world constraints. (ii) Novel methodology and baseline: We propose MIME, a framework for effective model extraction under minimal information. Experiments show MIME consistently outperforms existing methods in this setting, establishing a new performance baseline. (iii) Security implications: MIME offers attackers a blueprint for stealthy, low-budget extraction, while providing defenders a benchmark that highlights the limitations of simple query monitoring.

## 2 Preliminaries and Problem Formulation

**Preliminaries.** We consider a graph $G_{\text{full}} = (V_{\text{full}}, E_{\text{full}})$ with features $X_{\text{full}} \in \mathbb{R}^{|V_{\text{full}}| \times d}$, which is hidden from the attacker. A Graph Neural Network (GNN), $f(A, X; \theta)$, learns node representations via message passing (Gilmer et al., 2017). Our work focuses on transductive node classification, where a model predicts labels for nodes within a given graph (Kipf & Welling, 2017). Let $C$ be the number of classes. Our "cold start" strategy uses Deep Graph Infomax (DGI), an unsupervised method that learns node embeddings by maximizing mutual information between local and global representations, requiring no label information (Velickovic et al., 2019). For simplicity, we denote the victim's output vector for a node $v$ as $f_v(v) \in \mathbb{R}^C$.

**Attack Setting.** Unlike existing GNN extraction attacks that assume access to shadow datasets or the victim's data distribution (Zhuang et al., 2024), our work focuses on a more realistic, strict black-box setting. The attacker operates in a strict black-box environment. Their knowledge is confined to an unlabeled, induced subgraph $G_{\text{sub}} = (V_{\text{sub}}, E_{\text{sub}})$, including its node features $X_{\text{sub}}$ and local adjacency matrix $A_{\text{sub}}$. The attacker has no initial labels and is entirely unaware of the global graph structure outside of $G_{\text{sub}}$. The architecture and parameters of the victim model $f_v$ are unknown.

The attacker interacts with the victim model's API under two critical constraints often imposed in MLaaS settings (Guan et al., 2024): (1) a small total query budget $B \ll |V_{\text{sub}}|$, and (2) hard-label feedback, where each query for a node $v \in V_{\text{sub}}$ returns only the predicted class label, denoted $y_v^{\text{victim}}$. Concretely, the service predicts on the full hidden graph and returns the hard label

$$y_v^{\text{victim}} = \arg\max_{c \in \{1,...,C\}} f_v(v)_c,$$

while the attacker can only submit node identifiers (or their handles within $G_{\text{sub}}$). The attacker's goal is to train a surrogate model $f_s$ that functionally mimics the victim model $f_v$.

**Evaluation Metrics vs. Attacker Knowledge.** Success is measured by two standard metrics on a held-out test set $V_{\text{test}}$: Accuracy (agreement with ground-truth labels) and Fidelity (agreement with $f_v$ predictions) (Jagielski et al., 2020). Accuracy relies on ground-truth labels and is only used for offline research evaluation (not available to the attacker), whereas Fidelity is measurable from the attacker's interaction with $f_v$.

Formally, let $y_v^{\text{true}}$ be the ground-truth label for a node $v$. The metrics are defined as:

$$\text{Accuracy} = \frac{1}{|V_{\text{test}}|} \sum_{v \in V_{\text{test}}} \mathbf{1}\Big\{\arg\max_c f_s(v)_c = y_v^{\text{true}}\Big\}, \tag{1}$$

$$\text{Fidelity} = \frac{1}{|V_{\text{test}}|} \sum_{v \in V_{\text{test}}} \mathbf{1}\Big\{\arg\max_c f_s(v)_c = \arg\max_c f_v(v)_c\Big\}. \tag{2}$$

Here, $\mathbf{1}\{\cdot\}$ denotes the indicator function.

**Problem Formulation.** We first formalize the constraints governing our attack setting.

**Assumption 1. Attacker Constraints.** *Let the victim model be an unknown GNN $f_v$. An adversary is given access to an induced subgraph $G_{\text{sub}} \subset G_{\text{full}}$ and a query oracle $\mathcal{O}(\cdot)$ with a total budget $B$. The oracle is constrained to providing only hard-label feedback ($\mathcal{O}(v) \to y_v^{victim} = \arg\max_c f_v(v)_c$), exposes no internal model states, and provides no ground-truth labels $\{y_v^{true}\}$.*

With these constraints established, we can now formally define the model extraction task.

**Definition 1. Model Extraction Task.** *Given $G_{\text{sub}}$ and budget $B$, the task involves selecting a query set $Q \subseteq V_{\text{sub}}$ (with $|Q| \leq B$), acquiring a labeled set $Y_Q = \{\mathcal{O}(v) \mid v \in Q\}$ by querying the oracle, and subsequently training a surrogate model $f_s$ on this data. The objective is to achieve high Fidelity, $f_s \approx f_v$, on unseen nodes.*

This definition of the task naturally leads to a budgeted optimization problem focused on maximizing a utility function $\mathcal{J}$.

**Problem 1. Budgeted Optimization.** *With an offline utility $\mathcal{J}$ combining Accuracy and Fidelity, the goal is to find an optimal query set $Q^\star$ that maximizes this utility after training:*

$$Q^\star = \arg\max_{Q: |Q| \leq B} \mathcal{J}\Big(f_s(\cdot; \theta_s^\star(Q))\Big), \quad \text{where} \quad \theta_s^\star(Q) \in \arg\min_{\theta_s} \mathcal{L}_{\text{sup}}(Q, Y_Q; \theta_s) + \lambda \mathcal{R}_{\text{graph}}.$$

This bi-level nature underscores the core difficulty of the "Cold Start in the Dark" setting. Unlike standard active learning where initial labels might guide the outer loop, here the adversary lacks ground-truth seeds. Consequently, the outer selection depends heavily on the inner surrogate's uncertainty estimates, which are themselves unstable due to data sparsity. This mutual dependency necessitates the iterative, feedback-driven framework we propose in MIME.

## 3 METHODOLOGY

Our model extraction framework is designed for a realistic black-box setting defined by four key constraints: (i) data scarcity (an unlabeled subgraph only), (ii) hard-label feedback, (iii) a restrictive query budget, and (iv) no initial ground-truth labels.

### 3.1 OVERVIEW OF THE ATTACK FRAMEWORK

Our attack follows an iterative, three-phase process, as illustrated in Figure 1 and detailed in Appendix Algorithm 1. This process trains a surrogate model $f_s$ from zero initial labels using a small query budget. It begins by addressing the cold start problem with unsupervised representation learning for initial node selection. The framework then enters an active learning loop, selecting uncertain yet diverse nodes for querying through a sequential filter. The surrogate is retrained after each round using a loss function with adaptive, topology-aware regularization. Once the budget is exhausted, a final self-training step improves the model at no additional cost.

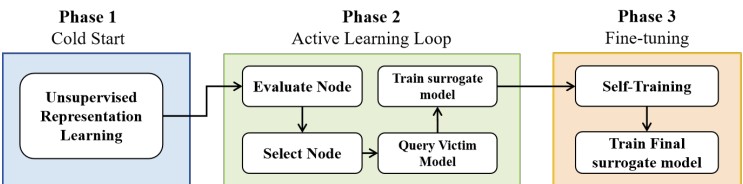

Figure 1: The three-phase workflow of MIME, progressing from a DGI-based cold start, through an active learning loop, to a final fine-tuning phase.

### 3.2 UNSUPERVISED PRE-TRAINING FOR COLD START INITIALIZATION

To address the cold start challenge where no initial labels exist, we employ Deep Graph Infomax (DGI) to learn structurally-aware node embeddings. Minimizing the DGI objective, which maximizes a lower bound on mutual information between local node patches and the global graph summary, is formulated as:

$$\mathcal{L}_{\mathrm{DGI}}(\theta_s) = -\sum_{v \in V_{\mathrm{sub}}} \Big[ \log \sigma(D(h_v, s)) + \log \sigma(-D(\tilde{h}_v, s)) \Big]. \tag{3}$$

In equation 3, an encoder $g_\theta$ generates node embeddings $h_v$, and the surrogate $f_s$ adds a classification head. The global summary $s$ is the mean of all node embeddings ($s = \frac{1}{|V_{\mathrm{sub}}|} \sum_{u \in V_{\mathrm{sub}}} h_u$), and negative samples $\tilde{h}_v$ are created by corrupting input features via row-wise shuffling. $D(\cdot, \cdot)$ is a bilinear discriminator and $\sigma(\cdot)$ is the sigmoid function. To measure dissimilarity within this learned latent space during the subsequent selection process, we utilize the angular distance $d(u, v) = \arccos(\mathrm{clip}(\frac{\langle u, v \rangle}{\|u\|_2 \|v\|_2}, -1, 1))$. This metric is derived from cosine similarity, satisfies the triangle inequality, and is empirically robust in high-dimensional embedding spaces.

**Adaptive Hybrid Initialization.** Relying solely on static embeddings for the initial query set $Q_0$ can be deterministic and potentially biased towards dominant structural communities. To enhance robustness and coverage, particularly for large-scale graphs, we introduce an adaptive hybrid selection strategy. Let $k = |Q_0|$ be the total size of the initial batch allowed by the budget. We decompose this budget into a random component $k_{\mathrm{rand}}$ and a structural component $k_{\mathrm{struct}}$ based on a dynamic mixing coefficient $\epsilon \in [0, 1]$:

$$k_{\mathrm{rand}} = \lfloor \epsilon \cdot k \rfloor, \quad k_{\mathrm{struct}} = k - k_{\mathrm{rand}}. \tag{4}$$

The mixing coefficient $\epsilon$ decays as the relative budget increases, reflecting the intuition that stochastic exploration is crucial when the budget is scarce relative to the graph size. The $k_{\mathrm{rand}}$ nodes are sampled uniformly at random to ensure unbiased global coverage. The remaining $k_{\mathrm{struct}}$ nodes are selected via a farthest-first $k$-center algorithm on the DGI embeddings using the angular distance defined above, ensuring the initialization covers the geometric convex hull of the data manifold.

### 3.3 ITERATIVE QUERY STRATEGY: SEQUENTIAL FILTERING

**Rounds and Budget.** The total query budget is $B$. Queries are performed in rounds $\gamma = 1, \dots, \Gamma$, with a batch size of $q$ nodes per round. The total number of rounds is $\Gamma = \lceil (B - |Q_0|)/q \rceil$. We set $Q^{(0)} = Q_0$ and $Q^{(\gamma)} = \bigcup_{t=0}^{\gamma} Q_t$.

Given the strict query budget, each query must be maximally informative. Our selection mechanism adopts a sequential filtering pipeline: the current surrogate $f_s^{(\gamma-1)}$ first identifies a pool of highly uncertain nodes, and then a diverse, class-balanced batch is chosen from this pool.

**Stage 1: Uncertainty-Based Candidate Pooling.** For each unqueried node $v \in V_{\text{sub}} \setminus Q^{(\gamma-1)}$, we compute a composite uncertainty score $U(v)$ from the output distribution $p_v$:

$$U(v) \;=\; w_{\text{ent}}\Big( -\sum_{c=1}^{C} p_{v,c} \log p_{v,c} \Big) \;+\; w_{\text{mar}}\Big( 1 - \big(p_v^{(1)} - p_v^{(2)}\big)\Big). \tag{5}$$

The weights are non-negative ($w_{\text{ent}} + w_{\text{mar}} = 1$). The rationale for this dual-term design is to capture complementary aspects of uncertainty: the entropy term measures global confusion across the entire output distribution, while the margin term specifically targets the decision boundaries between the top competing classes. By combining them, we mitigate the sampling bias that often arises from relying on a single heuristic, ensuring a more robust candidate pool $P_\gamma$ (top-$m_\gamma$ nodes, where $m_\gamma = \kappa q$).

**Stage 2: Diversity-Enforced Final Selection.** To avoid redundant queries within the candidate pool, we enforce diversity via a $k$-center objective in the dynamic embedding space of the current surrogate model. We select a batch $Q_\gamma$ of size $q$ by solving:

$$Q_\gamma^* \;=\; \arg \min_{\substack{Q \subseteq P_\gamma \\ |Q|=q}} \max_{v \in P_\gamma} \min_{u \in Q} d\big(h_v^{(\gamma-1)}, h_u^{(\gamma-1)}\big), \tag{6}$$

which is approximated by a greedy strategy.

Crucially, to prevent the "confirmation bias" typical of early-stage active learning—where the model disproportionately samples from classes it is already confident in—we impose a dynamic per-class quota $q_c$:

$$q_c \;=\; \max\Big(1, \Big\lceil \beta \cdot \frac{q}{C} \Big\rceil\Big), \qquad \forall\, c \in \{1, \dots, C\}, \quad \beta \in (0,1]. \tag{7}$$

This constraint forces the selection algorithm to explore under-represented classes, preventing the query budget from being exhausted on a single, potentially easy region of the graph. If the constraint is too restrictive, we gradually relax $\beta$ toward 1.

## 3.4 SURROGATE MODEL TRAINING AND REGULARIZATION

Training a robust surrogate from a sparse set of hard labels requires regularizing the learning process against overfitting. At each round $\gamma$, the model is trained by minimizing a composite loss:

$$\mathcal{L}_{\text{train}}(\theta_s) = \mathcal{L}_{\text{CE}}(Q^{(\gamma)}, Y_Q^{(\gamma)}) + \lambda_{\text{lap}}(\gamma)\mathcal{L}_{\text{lap}}. \tag{8}$$

The first term is the standard cross-entropy loss. The second term, $\mathcal{L}_{\text{lap}}$, is a node-level adaptive Graph Laplacian regularizer:

$$\mathcal{L}_{\text{lap}} = \frac{1}{|E_{\text{sub}}|} \sum_{(i,j) \in E_{\text{sub}}} \sqrt{w_i w_j} \left\| \frac{z_i}{\sqrt{\deg(i)+1}} - \frac{z_j}{\sqrt{\deg(j)+1}} \right\|_2^2. \tag{9}$$

This regularization term introduces a critical inductive bias based on the graph homophily assumption—that connected nodes likely share similar latent representations. By penalizing feature discrepancies across edges, $\mathcal{L}_{\text{lap}}$ effectively propagates label information from the sparse query set to unlabeled neighbors, smoothing the decision boundary along the data manifold and compensating for the scarcity of explicit supervision.

## 3.5 FINE-TUNING WITH SELF-TRAINING

Once the total query budget $B$ is exhausted, no further external information can be acquired. To fully exploit the learned model, we perform a final self-training step. We identify a set of high-confidence nodes $V_{\text{pseudo}}$ where the maximum prediction probability exceeds a threshold $\tau^\star$, and treat these predictions as pseudo-ground-truth. The model is fine-tuned by minimizing:

$$\mathcal{L}_{\text{final}} = \mathcal{L}_{\text{CE}}(Q^{(\Gamma)}, Y_Q^{(\Gamma)}) + \lambda_{\text{pseudo}}\, \mathcal{L}_{\text{CE}}(V_{\text{pseudo}}, Y_{\text{pseudo}}). \tag{10}$$

This phase is grounded in the cluster assumption, which posits that decision boundaries should pass through low-density regions. By reinforcing high-confidence predictions, we push the decision boundaries away from dense clusters, resolving remaining ambiguities and improving generalization at no additional query cost.

### 3.6 THEORETICAL ANALYSIS: A GEOMETRY-BASED GUARANTEE

We view querying as selecting a metric core-set in the embedding space. Let $h_v$ be a node embedding, $d(\cdot, \cdot)$ the angular distance, and $\ell(v)$ a per-node loss.

**Theorem 1** (Core-set (radius) bound). Assume $\ell$ is $\lambda$-Lipschitz w.r.t. $d$: $|\ell(v) - \ell(u)| \leq \lambda\, d(h_v, h_u)$. For any $Q \subseteq V_{\text{sub}}$, define the (covering) radius $\delta_Q := \max_{v \in V_{\text{sub}}} \min_{u \in Q} d(h_v, h_u)$, and let $\pi(v) \in \arg\min_{u \in Q} d(h_v, h_u)$ be the nearest queried representative. Then

$$\left| \frac{1}{|V_{\text{sub}}|} \sum_{v \in V_{\text{sub}}} \ell(v) - \frac{1}{|V_{\text{sub}}|} \sum_{v \in V_{\text{sub}}} \ell(\pi(v)) \right| \leq \lambda\, \delta_Q. \tag{11}$$

**Implication.** Greedy $k$-center (with uncertainty prefiltering) reduces $\delta_Q$ up to a constant factor (Gonzalez, 1985); moreover $Q^{(\gamma)} \subseteq Q^{(\gamma+1)}$ implies $\delta$ decreases monotonically (App. C.2), so Eq. equation 11 tightens with budget.

## 4 EXPERIMENTAL EVALUATION

In this section, we systematically evaluate MIME. Our experiments are designed to address four central research questions regarding efficiency, robustness, and internal design: **RQ1: Holistic Efficiency:** Does MIME achieve high extraction performance under tight budgets with reasonable computational overhead? **RQ2: Prior Sensitivity:** How does the size of the initial query pool impact performance? **RQ3: Robustness:** Is MIME effective across different victim architectures? **RQ4: Component Analysis:** What are the contributions of the framework's individual modules?

### 4.1 EXPERIMENTAL SETUP

**Datasets.** We evaluate MIME on five diverse benchmarks: Coauthor-CS (CoCS), Amazon-Computers (AmzC), CoraFull, ogbn-arxiv (Arxiv), and a 200,000-node subgraph of ogbn-products (Products). These datasets span a broad spectrum of scales and edge formation logics, ranging from citation homophily to functional complementarity in co-purchase networks, allowing us to assess performance on implicitly heterogeneous structures. Detailed statistics are in Appendix Table 4.

**Attack Scenario and Architectures.** We simulate a strict black-box attack where the attacker observes an unlabeled subgraph sampled from the victim's distribution. This prior subgraph is fixed at 10% for standard evaluations and varied from 1% to 10% for sensitivity analysis (RQ2). To verify robustness (RQ3), we attack three distinct pre-trained victim architectures: GCN (default), GraphSAGE, and GAT. All models utilize consistent hyperparameters to ensure fair evaluation (see Appendix Table 5).

**Baselines and Fair Comparison Protocol.** We conduct a comparative analysis against four representative methods spanning passive heuristics and active learning paradigms. Random serves as the standard passive baseline employing uniform sampling. To represent graph active learning strategies, we evaluate AGE (Cai et al., 2017), which aggregates centrality, density, and uncertainty metrics, and CEGA (Wang et al., 2025), which employs a dynamic weighting mechanism for query selection. Additionally, we include the Realistic Attack (Guan et al., 2024), a recent passive approach relying on structure reconstruction. To ensure the isolation of algorithmic efficacy, this evaluation adheres to a stringent fairness protocol: all methods operate under Cold Start (zero initial labeled seeds) and Hard Labels Only constraints. Furthermore, all surrogates share an Identical GCN Architecture and are subjected to the exact same Fixed Budget limits. The granular configurations and specific methodological mechanisms underpinning this rigorous comparison are detailed in Appendix Table 6.

**Metrics and Reproducibility.** Success is quantified by Effectiveness (Accuracy and Fidelity on a held-out test set) and Cost (Query Budget and Computational Overhead). Crucially, to ensure

statistical significance, all reported results represent the mean and standard deviation calculated over 5 independent runs with different random seeds.

## 4.2 EFFICIENCY ANALYSIS: IMPACT OF QUERY BUDGETS

To address RQ1, we evaluated MIME against baselines across four distinct total query budgets (5C to 20C). The comprehensive Accuracy and Fidelity scores are detailed in Table 1. A key trend emerges: MIME's performance trajectory is notably steeper and more consistent than competitors, establishing a robust overall advantage. Critically, this evaluation highlights the immediate computational failure of the Realistic Attack (denoted by –) on the large Arxiv and Products datasets. This confirms that structural augmentation methods designed for small graphs face insurmountable computational barriers when scaled, validating the necessity of a resource-aware framework like MIME. Detailed discussion on the complexity is provided in Appendix Section E.4.

**Superior Accuracy and Consistent Performance.** MIME demonstrates robust superiority in achieving high practical utility (Accuracy) by securing the highest mean score across virtually all tested scenarios and budget regimes. The only exception occurs at the minimum budget (5C) for the highly multi-class CoraFull dataset ($C = 70$). This marginal initial deficit is attributable to the large class complexity, which slightly challenges DGI's cold-start phase. However, MIME quickly establishes and maintains its lead in all subsequent budget regimes.

**Fidelity and Algorithmic Alignment.** MIME also maintains a dominant overall lead in Fidelity. It is occasionally surpassed by the Random baseline in this metric on the AmazonC and Arxiv datasets. This phenomenon highlights that Random sampling can, by coincidence, capture a set of labels highly consistent with the victim's *simple* decision boundaries. However, MIME's reliance on structured active learning ensures a superior alignment with the underlying data manifold, resulting in a more generalizable and higher-utility surrogate model. This validates MIME's exceptional query efficiency for robust IP extraction.

**Computational Overhead Analysis.** Figure 2 compares the total wall-clock time for all methods at the maximum budget (20C). The y-axis uses a logarithmic scale to accommodate the vast differences in magnitude. As shown, MIME (purple bars) consistently incurs a higher time cost compared to simple baselines like Random or CEGA. This is an expected trade-off: unlike heuristic methods, MIME performs a computationally intensive cold-start pre-training (DGI) and multiple rounds of surrogate retraining to maximize information gain from each query. Despite being slower, MIME remains within a practical range (e.g., $< 400$ seconds even for the large Products dataset). In stark contrast, the Realistic Attack (orange bars), while fast on small graphs, exhibits ex-

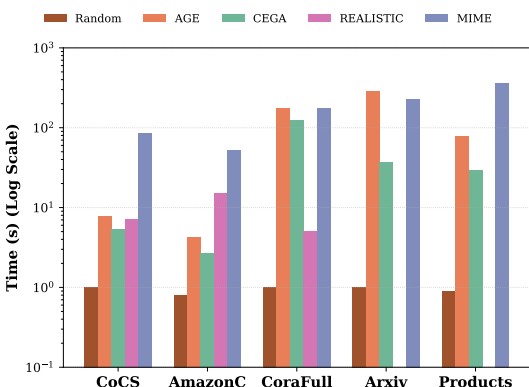

Figure 2: Computational time comparison at 20C budget (log scale). MIME incurs higher computational time costs compared to other methods

ponential time growth on large graphs due to global pairwise similarity checks, effectively timing out on Arxiv and Products. The additional time investment of MIME yields a disproportionately high return in Accuracy (Table 1), making it a worthwhile trade-off for adversaries prioritizing extraction quality over raw speed.

## 4.3 SENSITIVITY ANALYSIS: IMPACT OF PRIOR KNOWLEDGE

To address RQ2, we assess the sensitivity of MIME to the initial information available to the attacker. We vary the size of the initial query pool (prior knowledge) from 1% to 10% of the total graph nodes. For brevity, Figure 3 illustrates the Accuracy trends on three representative datasets spanning different scales: CoCS (Small), CoraFull (Medium), and Products (Large). The complete results, including both Accuracy and Fidelity across all five datasets, are provided in **Appendix Figure 4**.

Table 1: Comparison of accuracy and fidelity across methods and total query budgets on five diverse datasets (CoCS, AmzC, CoraFull, Arxiv, Products). The best performance for each budget and dataset is highlighted in bold.

| Dataset | Method | Accuracy Query Budget | | | | Fidelity Query Budget | | | |
|---|---|---|---|---|---|---|---|---|---|
| | | 5C | 10C | 15C | 20C | 5C | 10C | 15C | 20C |
| CoCS | Random | 0.7665 ± 0.05 | 0.8306 ± 0.02 | 0.8722 ± 0.01 | 0.8783 ± 0.00 | 0.7822 ± 0.05 | 0.8491 ± 0.02 | 0.8941 ± 0.01 | 0.9016 ± 0.00 |
| | AGE | 0.5653 ± 0.01 | 0.5779 ± 0.01 | 0.6461 ± 0.02 | 0.7298 ± 0.02 | 0.5761 ± 0.01 | 0.5903 ± 0.01 | 0.6585 ± 0.02 | 0.7448 ± 0.02 |
| | CEGA | 0.5442 ± 0.03 | 0.5574 ± 0.00 | 0.5906 ± 0.01 | 0.6044 ± 0.01 | 0.5550 ± 0.04 | 0.5686 ± 0.01 | 0.6037 ± 0.01 | 0.6173 ± 0.01 |
| | REALISTIC | 0.7428 ± 0.03 | 0.8119 ± 0.01 | 0.8600 ± 0.00 | 0.8727 ± 0.00 | 0.7582 ± 0.04 | 0.8297 ± 0.02 | 0.8811 ± 0.00 | 0.8956 ± 0.01 |
| | MIME | **0.8528 ± 0.02** | **0.8815 ± 0.01** | **0.9020 ± 0.01** | **0.9072 ± 0.01** | **0.8620 ± 0.02** | **0.8922 ± 0.01** | **0.9140 ± 0.01** | **0.9185 ± 0.01** |
| AmazonC | Random | 0.6773 ± 0.03 | 0.7030 ± 0.02 | 0.7221 ± 0.02 | 0.7286 ± 0.02 | 0.8329 ± 0.03 | 0.8568 ± 0.03 | 0.8749 ± 0.02 | 0.8929 ± 0.01 |
| | AGE | 0.5863 ± 0.06 | 0.6841 ± 0.01 | 0.6980 ± 0.02 | 0.7053 ± 0.01 | 0.7112 ± 0.09 | 0.8372 ± 0.02 | 0.8623 ± 0.03 | 0.8665 ± 0.01 |
| | CEGA | 0.6508 ± 0.01 | 0.7001 ± 0.02 | 0.7030 ± 0.02 | 0.7073 ± 0.01 | 0.7897 ± 0.01 | **0.8600 ± 0.02** | 0.8676 ± 0.03 | 0.8671 ± 0.02 |
| | REALISTIC | 0.6763 ± 0.01 | 0.6960 ± 0.02 | 0.7242 ± 0.01 | 0.7284 ± 0.01 | **0.8349 ± 0.01** | 0.8586 ± 0.02 | **0.8889 ± 0.01** | **0.8974 ± 0.01** |
| | MIME | **0.7366 ± 0.03** | **0.7312 ± 0.03** | **0.7514 ± 0.01** | **0.7668 ± 0.01** | 0.8137 ± 0.02 | 0.8055 ± 0.03 | 0.8308 ± 0.01 | 0.8541 ± 0.01 |
| CoraFull | Random | **0.3866 ± 0.01** | 0.4648 ± 0.01 | 0.5104 ± 0.01 | 0.5308 ± 0.01 | **0.4310 ± 0.01** | 0.5258 ± 0.02 | 0.5803 ± 0.01 | 0.6088 ± 0.00 |
| | AGE | 0.2166 ± 0.00 | 0.3549 ± 0.01 | 0.4684 ± 0.00 | 0.5193 ± 0.00 | 0.2527 ± 0.00 | 0.4042 ± 0.01 | 0.5305 ± 0.01 | 0.5939 ± 0.01 |
| | CEGA | 0.2250 ± 0.01 | 0.3803 ± 0.01 | 0.4816 ± 0.01 | 0.5188 ± 0.00 | 0.2602 ± 0.01 | 0.4316 ± 0.01 | 0.5471 ± 0.01 | 0.5860 ± 0.01 |
| | REALISTIC | 0.3285 ± 0.01 | 0.4127 ± 0.01 | 0.4681 ± 0.01 | 0.4987 ± 0.00 | 0.3692 ± 0.01 | 0.4694 ± 0.01 | 0.5342 ± 0.00 | 0.5714 ± 0.00 |
| | MIME | 0.3688 ± 0.02 | **0.4892 ± 0.00** | **0.5374 ± 0.00** | **0.5634 ± 0.00** | 0.3939 ± 0.03 | **0.5264 ± 0.00** | **0.5860 ± 0.00** | **0.6163 ± 0.00** |
| Arxiv | Random | 0.4683 ± 0.01 | 0.4910 ± 0.00 | 0.4959 ± 0.01 | 0.4982 ± 0.01 | **0.7369 ± 0.01** | **0.7838 ± 0.01** | **0.7943 ± 0.01** | **0.7955 ± 0.01** |
| | AGE | 0.4096 ± 0.03 | 0.4428 ± 0.01 | 0.4624 ± 0.01 | 0.4689 ± 0.00 | 0.6253 ± 0.04 | 0.6976 ± 0.02 | 0.7249 ± 0.01 | 0.7433 ± 0.01 |
| | CEGA | 0.3887 ± 0.02 | 0.4464 ± 0.01 | 0.4555 ± 0.01 | 0.4695 ± 0.01 | 0.6023 ± 0.04 | 0.7079 ± 0.02 | 0.7221 ± 0.02 | 0.7524 ± 0.02 |
| | REALISTIC | – | – | – | – | – | – | – | – |
| | MIME | **0.4952 ± 0.01** | **0.5242 ± 0.00** | **0.5403 ± 0.00** | **0.5408 ± 0.01** | 0.6607 ± 0.01 | 0.7111 ± 0.00 | 0.7351 ± 0.00 | 0.7385 ± 0.02 |
| Products | Random | 0.6601 ± 0.01 | 0.7158 ± 0.01 | 0.7401 ± 0.01 | 0.7566 ± 0.01 | 0.7259 ± 0.01 | 0.7926 ± 0.01 | 0.8209 ± 0.01 | 0.8411 ± 0.01 |
| | AGE | 0.5710 ± 0.01 | 0.6202 ± 0.03 | 0.6449 ± 0.02 | 0.6624 ± 0.01 | 0.6244 ± 0.01 | 0.6831 ± 0.03 | 0.7121 ± 0.02 | 0.7327 ± 0.01 |
| | CEGA | 0.5839 ± 0.02 | 0.6263 ± 0.03 | 0.6498 ± 0.02 | 0.6676 ± 0.01 | 0.6387 ± 0.02 | 0.6898 ± 0.03 | 0.7186 ± 0.02 | 0.7394 ± 0.01 |
| | REALISTIC | – | – | – | – | – | – | – | – |
| | MIME | **0.7194 ± 0.01** | **0.7803 ± 0.00** | **0.7939 ± 0.01** | **0.8172 ± 0.00** | **0.7716 ± 0.01** | **0.8413 ± 0.00** | **0.8572 ± 0.01** | **0.8844 ± 0.01** |

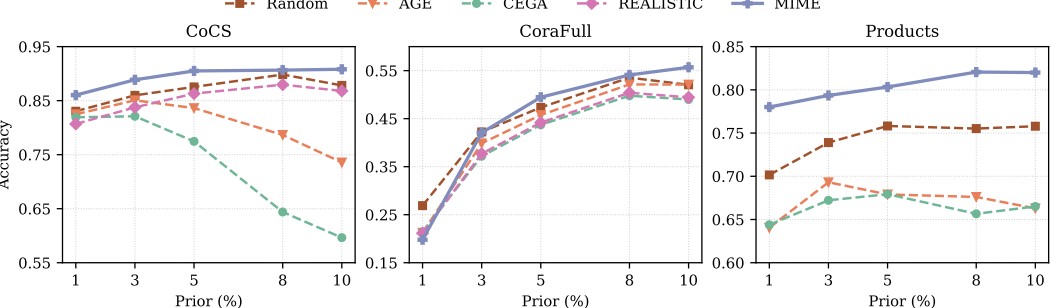

Figure 3: Accuracy of different methods under varying prior knowledge sizes (1%–10%) across three representative benchmark datasets: CoCS, CoraFull, and Products. MIME (purple line) demonstrates superior robustness, consistently outperforming baselines particularly in the low-prior regime (1-3%) and maintaining the lead as prior knowledge increases.

**Robustness to Prior Scarcity.** The results in Figure 3 demonstrate MIME's exceptional robustness. Even with minimal prior knowledge (1%–3%), MIME establishes a clear performance advantage over baselines, effectively overcoming the cold-start challenge where other active learning methods (like AGE and CEGA) struggle to initialize. As the prior size increases to 10%, MIME maintains its lead, exhibiting a stable and monotonic improvement trajectory.

**Accuracy-Fidelity Trade-off.** As detailed in the full results (Appendix Figure 4), while MIME consistently dominates in Accuracy, its Fidelity scores are occasionally matched or slightly surpassed by the others on specific datasets. However, this gap is marginal. Crucially, MIME achieves this high Fidelity while delivering significantly higher Accuracy than others. This indicates that MIME does not merely memorize the victim's behavior but learns a generalizable decision boundary that is both faithful to the victim and accurate on the ground truth, making it a more potent attack vector.

## 4.4 ROBUSTNESS ANALYSIS: IMPACT OF VICTIM ARCHITECTURES

To answer RQ3, we evaluate the resilience of MIME when the target victim model's architecture is unknown and differs from the surrogate. We fix the surrogate as a GCN and attack three distinct victim architectures: GCN, GraphSAGE, and GAT, under a fixed budget of 20C. The extraction results are summarized in Table 2.

Table 2: MIME performance under 20C query budget across varying victim architectures. Shadow model is fixed as GCN.

| Victim | Metric | CoCS | AmzC | CoraFull | Arxiv | Products |
|--------|--------|------|------|----------|-------|----------|
| GCN | Acc | $0.9083 \pm 0.0025$ | $0.7678 \pm 0.0093$ | $0.5605 \pm 0.0085$ | $0.5395 \pm 0.0063$ | $0.8155 \pm 0.0052$ |
| SAGE | Acc | $0.9088 \pm 0.0046$ | $0.4717 \pm 0.0052$ | $0.5539 \pm 0.0042$ | $0.5411 \pm 0.0090$ | $0.8147 \pm 0.0029$ |
| GAT | Acc | $0.9146 \pm 0.0077$ | $0.8014 \pm 0.0242$ | $0.5468 \pm 0.0258$ | $0.5405 \pm 0.0214$ | $0.8165 \pm 0.0016$ |
| GCN | Fid | $0.9217 \pm 0.0017$ | $0.8493 \pm 0.0037$ | $0.6113 \pm 0.0077$ | $0.7334 \pm 0.0110$ | $0.8825 \pm 0.0064$ |
| SAGE | Fid | $0.9197 \pm 0.0041$ | $0.9455 \pm 0.0045$ | $0.5965 \pm 0.0040$ | $0.7159 \pm 0.0166$ | $0.8814 \pm 0.0048$ |
| GAT | Fid | $0.9311 \pm 0.0080$ | $0.8451 \pm 0.0314$ | $0.6002 \pm 0.0276$ | $0.6893 \pm 0.0311$ | $0.8896 \pm 0.0013$ |

**Architecture Agnostic Performance.** The results demonstrate that MIME maintains high stability and effectiveness regardless of the victim's internal mechanism. As shown in Table 2, the Accuracy and Fidelity scores across CoCS, CoraFull, Arxiv, and Products remain remarkably consistent. This confirms that MIME successfully extracts the generalizable decision logic of the target rather than overfitting to specific architectural artifacts.

**Analysis of Variations.** A noticeable fluctuation occurs on the AmzC dataset, particularly with the GraphSAGE victim. However, this is not a failure of the extraction framework. As detailed in Appendix Table 7, the GraphSAGE architecture itself performs poorly and unstably on Amazon-Computers, significantly lower than GAT or GCN. Consequently, MIME's extraction performance is naturally bounded by the quality of the victim it is trying to mimic. Notably, MIME's surrogate often achieves higher accuracy than the suboptimal SAGE victim itself, suggesting the surrogate (GCN) effectively regularizes the noisy supervision from the weak victim.

### 4.5 ABLATION STUDY: IMPACT OF FRAMEWORK COMPONENTS

To answer RQ4 and quantify each component's contribution, we conduct an ablation study using a leave-one-out methodology. This approach assesses the marginal impact of removing a single module from the full MIME framework. We evaluate six ablated configurations by individually removing: (i) DGI pre-training, (ii) diversity-based query selection, (iii) the class-balancing quota, (iv) the graph Laplacian regularizer, (v) the final self-training phase, and (vi) the use of dynamic embeddings (reverting to static ones). The complete results and detailed configuration descriptions are available in the Appendix Table 9.

**Component Necessity and Synergy.** The results in Table 3 confirm that each module is integral to MIME's performance. The complete framework achieves the highest accuracy on four out of five datasets (CoCS, AmzC, Cora, and Arxiv). Removing foundational components like DGI pre-training, the Laplacian regularizer, and the class quota consistently degrades performance, validating their necessity. While a minor exception is observed on the Products dataset where removing Self-Training yields a marginal gain, this module remains critical for significant improvements on others (e.g., CoCS and AmzC), demonstrating the overall robustness of the integrated design.

Table 3: Ablation study results on accuracy across five benchmark datasets. The best performance for each dataset is highlighted in bold.

| Ablation | CoCS | AmzC | Cora | Arxiv | Products |
|----------|------|------|------|-------|----------|
| MIME | $\mathbf{0.9059 \pm 0.00}$ | $\mathbf{0.7668 \pm 0.01}$ | $\mathbf{0.5547 \pm 0.00}$ | $\mathbf{0.5591 \pm 0.01}$ | $0.8192 \pm 0.00$ |
| No DGI | $0.8989 \pm 0.01$ | $0.7328 \pm 0.02$ | $0.5513 \pm 0.00$ | $0.5508 \pm 0.00$ | $0.8198 \pm 0.00$ |
| No Diversity | $0.9047 \pm 0.00$ | $0.7373 \pm 0.01$ | $0.5505 \pm 0.00$ | $0.5582 \pm 0.00$ | $0.8130 \pm 0.00$ |
| No Quota | $0.8970 \pm 0.00$ | $0.7385 \pm 0.03$ | $0.5517 \pm 0.00$ | $0.5497 \pm 0.00$ | $0.8192 \pm 0.00$ |
| No Laplacian | $0.9005 \pm 0.01$ | $0.7415 \pm 0.01$ | $0.5500 \pm 0.00$ | $0.5526 \pm 0.00$ | $0.8174 \pm 0.00$ |
| No SelfTrain | $0.8935 \pm 0.00$ | $0.7555 \pm 0.01$ | $0.5448 \pm 0.00$ | $0.5501 \pm 0.00$ | $\mathbf{0.8226 \pm 0.00}$ |
| Static Embeddings | $0.9047 \pm 0.00$ | $0.7172 \pm 0.01$ | $0.5492 \pm 0.00$ | $0.5562 \pm 0.01$ | $0.8175 \pm 0.00$ |

## 5 RELATED WORK

**Dependency on Unrealistic Priors: Soft Labels and Permissive Budgets.** Most GNN model extraction attacks rely on favorable conditions absent in secured MLaaS. A dominant stream necessitates **soft-label feedback** for knowledge distillation (Wu et al., 2022), yet secure APIs typically restrict outputs to hard labels. Similarly, transfer-based approaches like Knockoff Nets(Orekondy et al., 2019) assume access to auxiliary datasets matching the victim's distribution. In our transductive setting, where the adversary observes a specific subgraph without external proxy graphs, such strategies are inapplicable or degenerate into random sampling due to the lack of queryable out-of-distribution inputs. To bypass data dependencies, data-free approaches like StealGNN (Zhuang et al., 2024) use generative models to synthesize queries. While theoretically viable without surrogate data, training a generator to map decision boundaries via hard labels is inherently query-intensive, often requiring tens of thousands of queries to estimate gradients. This violates the strict financial and rate-limiting constraints of real-world attacks. Our "Cold Start in the Dark" setting imposes a tight budget, rendering such generator-based strategies computationally infeasible.

**Initialization Bottleneck in Active Learning.** Query-efficient extraction parallels Graph Active Learning (GAL). Frameworks like AGE (Cai et al., 2017) and CEGA (Wang et al., 2025) select informative nodes via centrality and uncertainty. However, they suffer from a cold start problem: they require labeled seeds to compute initial metrics. MIME bridges this gap by integrating Self-Supervised Learning (Velickovic et al., 2019) as a zero-label initializer. By bootstrapping structural embeddings from topology alone, MIME enables effective extraction under stringent constraints, distinguishing it from prior data-dependent or budget-heavy approaches.

## 6 CONCLUSION

In this paper, we formally defined and addressed the "Cold Start in the Dark" problem, a stringent yet realistic model extraction scenario characterized by minimal initial information, hard-label feedback, and tight query budgets. To navigate this hostile environment, we introduced MIME, a framework that bridges the gap between theoretical vulnerability and practical exploitability. By bootstrapping from unsupervised topological knowledge and employing a geometry-aware active learning strategy, MIME effectively overcomes the lack of initial seed labels. Our extensive evaluation confirms that MIME consistently outperforms state-of-the-art baselines in both Accuracy and Fidelity across most diverse datasets and victim architectures. Furthermore, we provided a theoretical generalization bound based on the core-set radius, offering formal justification for the effectiveness of our diversity-driven query selection.

**Implications for defenses.** Our results highlight practical implications for defending MLaaS against extraction. First, to counter MIME's reliance on structural coverage, providers should reduce the attack surface by restricting neighborhood visibility and enforcing access controls that prevent the assembly of representative subgraphs. Second, since extraction exploits topological smoothness, defenders should employ calibrated output obfuscation, such as controlled label randomization for suspicious patterns or selective disclosure at decision boundaries. Finally, detection systems must evolve to stateful anomaly detection, auditing query streams for atypical coverage patterns or repeated boundary probing, triggering adaptive throttling when such stealthy signatures emerge.

**Limitations and future work.** This work focuses on transductive node classification with a fixed attacker-visible induced subgraph, and extending the framework to the inductive regime is an important next step. Future work also includes expanding to other tasks such as link prediction and graph-level prediction, evaluating robustness under graph families with strong heterophily or distribution shift between the attacker's observed subgraph and the victim's training distribution, and studying temporal or evolving graphs where access patterns and topology change over time. Another direction is to incorporate stronger operational constraints that reflect production settings, including per-query costs, delayed or stochastic responses, and explicit stealth constraints under active defenders. Finally, a systematic defense benchmark that evaluates detection and mitigation strategies under a shared protocol aligned with Cold Start in the Dark would help translate these insights into deployable safeguards.

ETHICS STATEMENT

We acknowledge that this work details a model extraction attack, a methodology with potential for malicious use. Our primary motivation is defensive: by developing and analyzing a more realistic and efficient attack vector, we aim to highlight critical vulnerabilities in current GNN-based MLaaS platforms. This research is intended to serve as a benchmark for the security community, enabling the development and evaluation of more robust defense mechanisms against such threats. The described methods were developed in a controlled, simulated environment. We believe that transparently discussing these vulnerabilities is crucial for motivating and informing the creation of stronger security protocols. Our work adheres to the principles of responsible research by focusing on the security implications and providing insights for defenders.

REPRODUCIBILITY STATEMENT

We have made every effort to ensure the reproducibility of our results. A detailed description of our proposed framework, MIME, including the unsupervised pre-training, iterative query strategy, and surrogate model training, is provided in Section 3. The full algorithm is presented in Algorithm 1 in the appendix. Our complete experimental setup, including dataset descriptions, data partitioning, and victim model training protocols, is detailed in Section 4.1. The specific hyperparameters used for all experiments are also listed in the appendix. All datasets used (CoCS, CoP, AmzC, AmzP, and Cora) are publicly available benchmarks. To facilitate full reproduction of our findings, we provide our source code, including the implementation of our method and all baselines, in the supplementary materials.

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

APPENDIX

## A  SUPPLEMENTARY INSTRUCTIONS FOR FORMULAS

**K-center Objective Details** The k-center objective seeks to select a subset of points $Q$ from a larger set $P_\gamma$ that minimizes the maximum distance from any point in $P_\gamma$ to its nearest point in $Q$. This is formally expressed as:

$$\phi(Q) = \max_{v \in P_\gamma} \min_{u \in Q} d\big(h_v^{(\gamma-1)}, h_u^{(\gamma-1)}\big), \tag{12}$$

where $d(\cdot, \cdot)$ is the distance metric. The farthest-first greedy algorithm provides a 2-approximation for this NP-hard problem.

**Laplacian Regularization Schedule: Intuition** The regularization strength $\lambda_{\text{lap}}(\gamma)$ is not static. An effective schedule starts with a small $\lambda_{\text{lap}}$ when the number of queried labels is low to avoid over-smoothing based on a potentially biased initial surrogate model. As more labels are acquired and the model becomes more confident, $\lambda_{\text{lap}}$ can be increased to enforce greater smoothness and improve generalization. For instance, a simple schedule could be $\lambda_{\text{lap}}(\gamma) = \lambda_0 \cdot \min(1, |Q^{(\gamma)}|/B_{thresh})$, where $\lambda_0$ is the base regularization strength and $B_{thresh}$ is a budget threshold after which the regularization is fully active. This adaptive approach prevents early, aggressive regularization from washing out the learning signal from the few available labels.

## B  NOTATION TABLE

| Symbol | Description |
|--------|-------------|
| $G_{\text{full}} = (V_{\text{full}}, E_{\text{full}})$ | Full (hidden) graph of the victim model |
| $G_{\text{sub}} = (V_{\text{sub}}, E_{\text{sub}})$ | Induced subgraph observed by the attacker |
| $N, E$ | Number of nodes and edges in $G_{\text{sub}}$, i.e., $N = |V_{\text{sub}}|, E = |E_{\text{sub}}|$ |
| $A_{\text{sub}}, X_{\text{sub}}$ | Adjacency and feature matrix on $G_{\text{sub}}$ |
| $d, d_e$ | Input feature dimension and dynamic embedding dimension |
| $C$ | Number of classes |
| $cc(v)$ | Local clustering coefficient of node $v$ |
| $f_v, f_s$ | Victim and surrogate GNNs; $f_v(v) \in \mathbb{R}^C$ is victim's output vector |
| $y_v^{\text{victim}}$ | Hard-label returned by the API: $y_v^{\text{victim}} = \arg\max_c f_v(v)_c$ |
| $B$ | Total query budget (issued in batches) |
| $q$ | Batch size per round (also $|Q_0|$ if $B \geq q$) |
| $\Gamma$ | Number of rounds: $\Gamma = \lceil (B - |Q_0|)/q \rceil$ |
| $H^{(0)} = \{h_v^{(0)}\}$ | DGI pre-trained (static) embeddings (cold start only) |
| $h_v^{(\gamma)}$ | Round-$\gamma$ embedding from $f_s^{(\gamma)}$ (dynamic) |
| $Q^{(\gamma)}$ | Queried set up to round $\gamma$; $Q^{(0)} = Q_0$ |
| $\kappa$ | Pool factor in candidate size $m_\gamma = \kappa q$ |
| $\beta$ | Per-class cap factor (Eq. equation 7) |
| $U(v)$ | Uncertainty score (entropy+margin) in Eq. equation 5 |
| $d(\cdot, \cdot)$ | Angular distance (cosine-induced metric) used in $k - center$ |
| $z_v, p_v$ | Logits and softmax probability for node $v$ |
| $\lambda_{\text{lap}}$ | Laplacian regularization weight (Eq. equation 8) |
| $\tau^\star$ | Confidence threshold for pseudo-labels in self-training |
| $\lambda_{\text{pseudo}}$ | Weight for pseudo-label loss (Eq. equation 10) |
| $T_{\text{vic}}, T_{\text{DGI}}, T_{\text{tr}}$ | Victim training, DGI pre-training, and per-round training epochs |

## C  PROOFS AND JUSTIFICATIONS

### C.1  ADDITIONAL NOTATION (FOR PROOFS)

Let $P$ be any non-empty finite set of nodes (in our use, $P = V_{\text{sub}}$). Let $Q$ be a non-empty set of queried representatives. Given embeddings $\{h_v\}$ and a distance function $d(\cdot, \cdot)$ on the embedding space, define the *covering radius* of $Q$ over $P$ as

$$\delta_Q(P) := \max_{v \in P} \min_{u \in Q} d(h_v, h_u). \tag{13}$$

For each $v \in P$, define a nearest-representative mapping

$$\pi_Q(v) \in \arg \min_{u \in Q} d(h_v, h_u), \tag{14}$$

with ties broken arbitrarily but deterministically.

**Well-posedness details.**  First, our arguments only require that $d(\cdot, \cdot)$ is non-negative and that $\min_{u \in Q} d(h_v, h_u)$ is well-defined; in particular, $d$ may be a metric or a pseudo-metric (allowing distinct points at zero distance), as symmetry/triangle inequality are never invoked in the proofs. Second, we explicitly assume $Q \neq \emptyset$; otherwise $\min_{u \in Q}$ is undefined and $\delta_Q(P)$ does not exist. Third, the tie-breaking rule in Eq. equation 14 does not affect any of our bounds, since all inequalities depend only on the value $\min_{u \in Q} d(h_v, h_u)$ (and we upper bound it by $\delta_Q(P)$), which is invariant to the choice of an argmin representative.

### C.2  MONOTONICITY OF THE COVERING RADIUS

We justify the monotone tightening statement used in §3.6.

**Lemma 1** (Monotone radius under superset queries). Let $P$ be non-empty and finite, and let $Q, Q'$ be non-empty query sets such that $Q \subseteq Q'$. Then the covering radius is non-increasing:

$$\delta_{Q'}(P) \leq \delta_Q(P). \tag{15}$$

*Proof.*  Fix any $v \in P$. Since $Q \subseteq Q'$, we have

$$\min_{u \in Q'} d(h_v, h_u) \leq \min_{u \in Q} d(h_v, h_u),$$

because the minimization in the left-hand side is taken over a superset. Taking $\max_{v \in P}$ on both sides yields

$$\max_{v \in P} \min_{u \in Q'} d(h_v, h_u) \leq \max_{v \in P} \min_{u \in Q} d(h_v, h_u),$$

which is exactly Eq. equation 15. $\square$

**Corollary (round-wise tightening).**  In MIME, the queried set grows monotonically across rounds: $Q^{(\gamma)} \subseteq Q^{(\gamma+1)}$. Applying Lemma 1 with $Q = Q^{(\gamma)}$ and $Q' = Q^{(\gamma+1)}$ gives $\delta_{Q^{(\gamma+1)}}(P) \leq \delta_{Q^{(\gamma)}}(P)$ for any fixed $P$ (in particular $P = V_{\text{sub}}$).

### C.3  PROOF DETAILS FOR THE GEOMETRY-BASED GUARANTEE (§3.6)

We provide a detailed justification for the geometric guarantee in §3.6. The key step is that Lipschitz losses can be transferred from any node $v$ to its nearest representative $\pi_Q(v)$ with an additive error controlled by the covering radius.

**Lemma 2** (Pointwise transfer via covering radius). Assume $\ell(\cdot)$ is $\lambda$-Lipschitz with respect to $d(\cdot, \cdot)$ on the embedding space, i.e., $|\ell(v) - \ell(u)| \leq \lambda \, d(h_v, h_u)$ for all $u, v$. Then for any non-empty $Q$ and any $v \in P$,

$$|\ell(v) - \ell(\pi_Q(v))| \leq \lambda \, \delta_Q(P). \tag{16}$$

*Proof.*  By definition of $\pi_Q(v)$ in Eq. equation 14,

$$d(h_v, h_{\pi_Q(v)}) = \min_{u \in Q} d(h_v, h_u) \leq \delta_Q(P),$$

where the last inequality follows from the definition of $\delta_Q(P)$ in Eq. equation 13. Applying $\lambda$-Lipschitzness yields

$$|\ell(v) - \ell(\pi_Q(v))| \ \leq \ \lambda\,d(h_v, h_{\pi_Q(v)}) \ \leq \ \lambda\,\delta_Q(P).$$

This proves Eq. equation 16. $\qquad\square$

**A precise population-to-coreset statement.** Define the partition induced by $\pi_Q$ as $S_u := \{v \in P : \pi_Q(v) = u\}$ for each $u \in Q$, and weights $w_u := |S_u|/|P|$ (so $\sum_{u \in Q} w_u = 1$). Then the population average loss is close to the *weighted* representative loss:

**Proposition 1** (Population-to-weighted-core-set bound). Under the assumptions of Lemma 2,

$$\left| \frac{1}{|P|} \sum_{v \in P} \ell(v) \ - \ \sum_{u \in Q} w_u\,\ell(u) \right| \ \leq \ \lambda\,\delta_Q(P). \tag{17}$$

*Proof.* Using the induced mapping $\pi_Q$ and re-grouping by representatives,

$$\sum_{v \in P} \ell(\pi_Q(v)) \ = \ \sum_{u \in Q} \sum_{v \in S_u} \ell(u) \ = \ \sum_{u \in Q} |S_u|\,\ell(u).$$

Therefore,

$$\frac{1}{|P|} \sum_{v \in P} \ell(\pi_Q(v)) \ = \ \sum_{u \in Q} \frac{|S_u|}{|P|} \ell(u) \ = \ \sum_{u \in Q} w_u\,\ell(u).$$

Now apply triangle inequality and Lemma 2:

$$\left| \frac{1}{|P|} \sum_{v \in P} \ell(v) \ - \ \sum_{u \in Q} w_u\,\ell(u) \right| = \left| \frac{1}{|P|} \sum_{v \in P} \big(\ell(v) - \ell(\pi_Q(v))\big) \right|$$

$$\leq \frac{1}{|P|} \sum_{v \in P} |\ell(v) - \ell(\pi_Q(v))|$$

$$\leq \frac{1}{|P|} \sum_{v \in P} \lambda\,\delta_Q(P) \ = \ \lambda\,\delta_Q(P).$$

This proves Eq. equation 17. $\qquad\square$

**Recovering the simplified (unweighted) form used in the main text.** If the induced cells $\{S_u\}$ are approximately balanced (i.e., $w_u \approx 1/|Q|$), then $\sum_{u \in Q} w_u \ell(u) \approx \frac{1}{|Q|} \sum_{u \in Q} \ell(u)$ and Eq. equation 17 yields the lightweight bound presented in Eq. (7) of §3.6. More generally, if $\ell(\cdot) \in [0, M]$ is bounded, then

$$\left| \sum_{u \in Q} w_u\,\ell(u) \ - \ \frac{1}{|Q|} \sum_{u \in Q} \ell(u) \right| \leq M \sum_{u \in Q} \left| w_u - \frac{1}{|Q|} \right|, \tag{18}$$

and combining Eq. equation 17 and Eq. equation 18 gives an explicit "coverage + imbalance" decomposition.

## C.4   *Necessity of DGI at Cold Start*

We recall the DGI objective from Eq. equation 3, which maximizes a mutual information bound between local embeddings $h_v$ and the global summary $s$ (Velickovic et al., 2019). This ensures $H^{(0)}$ encodes structural patterns (communities, roles) without labels. The initial query set $Q_0$ is chosen by farthest-first $k$-center (Eq. equation 12), which approximates the minimum covering radius within a factor 2 (Gonzalez, 1985). Thus, $Q_0$ provides diverse, label-free coverage—strictly better than random seeds. *Note:* As stated in §3.2, $H^{(0)}$ is used only at cold start.

### C.5 *Farthest-First (K-Center) for Diversity*

We recall the diversity objective in Eq. equation 6, defined over the candidate pool $P_\gamma$. Using the angular distance $d(\cdot, \cdot)$ (see §3.2), the greedy farthest-first heuristic achieves $\phi(Q_{\text{FF}}) \leq 2\phi(Q^*)$ (Gonzalez, 1985). This ensures constant-factor coverage, unlike $k$-means, which lacks worst-case guarantees.

### C.6 *Dynamic Embeddings for Diversity*

We recall the uncertainty score $U(v)$ from Eq. equation 5. After uncertainty filtering, diversity is applied in the *dynamic embedding space* $\{h_v^{(\gamma-1)}\}$ produced by $f_s^{(\gamma-1)}$. Because embeddings evolve as more labels are queried, this keeps diversity aligned with the surrogate's current decision geometry.

### C.7 *Necessity of Adaptive Laplacian Regularization*

We recall the training objective in Eq. equation 8, where $\mathcal{L}_{\text{lap}}$ (Eq. equation 9) penalizes discrepancies across edges with node-adaptive weights (Eq. equation **??**). A spectral view shows that regularization attenuates high-frequency modes, but excessive smoothing on sparse/heterophilous subgraphs is harmful. Hence $\lambda_{\text{lap}}$ is scheduled adaptively (Appendix A), gated by subgraph connectivity, ramped with labeled fraction, and modulated by homophily/spectral cues.

### C.8 *Convergence Trend*

Accuracy and Fidelity are the main evaluation metrics (Eq. equation 2); for intuition, define a surrogate–victim risk $R_\gamma = \mathbb{E}[\mathbf{1}\{\arg\max f_s^{(\gamma)} \neq \arg\max f_v\}]$. Uncertainty sampling reduces error-prone regions; diversity prevents redundancy. Thus, $R_\gamma$ is expected to *decrease in trend* as the queried set expands and coverage improves; meanwhile, we do *not* claim strict monotonic improvement of Accuracy/Fidelity under non-convex training and hard-label noise.

### C.9 *Computational Complexity*

We recall the round structure and symbols from §3.3. DGI pre-training costs $O(T_{\text{DGI}}Ed)$. Each round costs $O(Ed + NC + m_\gamma q d_e + T_{\text{tr}}Ed)$. The term $m_\gamma q d_e$ arises from the farthest-first selection, where $m_\gamma = \kappa q$; its complexity can be optimized in practice with data structures like heaps. Total cost:

$$O\Big(T_{\text{DGI}}Ed + \tfrac{B}{q}(T_{\text{tr}}Ed + NC + \kappa q^2 d_e)\Big).$$

Since $B \ll N$, training dominates. Space is $O(E + N(d + d_e + C))$, so MIME is polynomial-time feasible.

# D  FULL ALGORITHM SPECIFICATION

---

**Algorithm 1:** The proposed framework of MIME

---

**Initialization:** Pre-train DGI on $(A_{\text{sub}}, X_{\text{sub}})$ to obtain initial embeddings $H^{(0)}$.

    Select initial nodes $Q_0$ by farthest-first on $H^{(0)}$ with $|Q_0| = \min(q, B)$ from $V_{\text{sub}}$.

    Query victim API to get hard labels $Y_{Q_0}$, where $y_v^{\text{victim}} = \arg\max_c [f_v(A_{\text{full}}, X_{\text{full}})]_{v,c}$.

    Train the initial surrogate $f_s^{(0)}$ on $(Q_0, Y_{Q_0})$ by minimizing

    $\mathcal{L}_{\text{train}} = \mathcal{L}_{\text{CE}}(Q_0, Y_{Q_0}) + \lambda_{\text{lap}}(0)\mathcal{L}_{\text{lap}}$ for $T_{\text{tr}}$ epochs.

**for** *Cycle $\gamma$ from 1 to* $\Gamma = \lceil (B - |Q_0|)/q \rceil$ **do**

    **if** $|Q^{(\gamma-1)}| + q \leq B$ **then**

        Evaluate uncertainty score $U(v)$ for all $v \in V_{\text{sub}} \setminus Q^{(\gamma-1)}$ using Eq. equation 5.

        Build candidate pool $P_\gamma$ with top-$m_\gamma$ nodes, $m_\gamma = \kappa q$.

        Obtain dynamic embeddings $h_v^{(\gamma-1)}$ from $f_s^{(\gamma-1)}$.

        Select and query $q$ nodes $Q_\gamma$ via farthest-first on $P_\gamma$ with class cap (Eq. equation 7).[a]

        Obtain victim labels $Y_{Q_\gamma}$ and set $Q^{(\gamma)} = Q^{(\gamma-1)} \cup Q_\gamma$.

    **else**

        Set $Q^{(\gamma)} = Q^{(\gamma-1)}$.

    **end**

    Train $f_s^{(\gamma)}$ on $\{Q^{(\gamma)}, G_{\text{sub}}\}$ for $T_{\text{tr}}$ epochs by minimizing Eq. equation 8.

**end**

**Self-training:** Form $V_{\text{pseudo}} = \{v : \max_c p_{v,c} \geq \tau^\star\}$ with labels $Y_{\text{pseudo}}$ from $f_s^{(\Gamma)}$, and

                fine-tune with Eq. equation 10.

**Return:**

Queried nodes $\{Q^{(1)}, ..., Q^{(\Gamma)}\}$ and final surrogate $f_s^{\text{final}}$.

---

[a]If the remaining budget is less than $q$, this step selects only the number of remaining nodes allowed by the budget.

# E    SUPPLEMENTARY EXPERIMENTAL DETAILS

This section provides additional details regarding our experimental setup and presents supplementary results to ensure full reproducibility.

## E.1    DATASET STATISTICS

Table 4: Summary of dataset statistics and characteristics.

| Dataset | $N$ (Nodes) | $|E|$ (Edges) | $D$ (Features) | $C$ (Classes) |
|---|---|---|---|---|
| CoCS (Coauthor-CS) | 18,333 | 163,788 | 6,805 | 15 |
| AmzC (Amazon-Computers) | 13,752 | 491,722 | 767 | 10 |
| CoraFull | 19,793 | 126,842 | 8,710 | 70 |
| Arxiv (ogbn-arxiv) | 169,343 | 2,315,598 | 128 | 40 |
| Products (ogbn-products) | 200,000 | 11,105,650 | 100 | 42 |

## E.2    HYPERPARAMETER SETTINGS

Table 5: Comprehensive hyperparameter settings. We compare the configuration of the three victim architectures used for robustness analysis (GCN, GraphSAGE, GAT) against the Surrogate model used by MIME. Dash (–) indicates a parameter is not applicable.

| Parameter | Victim (GCN) | Victim (SAGE) | Victim (GAT) | Surrogate (MIME) |
|---|---|---|---|---|
| *Model Architecture* | | | | |
| Layers | 2 | 2 | 2 | 2 |
| Hidden Dimension | 16 | 16 | $8 \times 16$ (128 total) | 128 |
| Attention Heads | – | – | 8 (Layer 1) / 1 (Layer 2) | – |
| Aggregator | – | Mean | – | – |
| Dropout | 0.5 | 0.5 | 0.5 | 0.5 |
| *Optimization & Training* | | | | |
| Optimizer | Adam | Adam | Adam | Adam |
| Learning Rate | $1 \times 10^{-3}$ | $1 \times 10^{-3}$ | $1 \times 10^{-3}$ | $1 \times 10^{-3}$ |
| Weight Decay | $5 \times 10^{-4}$ | $5 \times 10^{-4}$ | $5 \times 10^{-4}$ | $5 \times 10^{-4}$ |
| Total Epochs | 1000 | 1000 | 1000 | Dynamic[†] |
| *MIME Specific Strategy* | | | | |
| Label Smoothing | – | – | – | 0.03 |
| DGI Pre-train Epochs | – | – | – | 300 |
| Epochs per Query Round | – | – | – | 100 |
| Final Fine-tuning Epochs | – | – | – | 200 |
| Laplacian Weight ($\lambda$) | – | – | – | $5 \times 10^{-4}$ |

Table 6: Detailed methodological comparison between MIME and baseline approaches. All methods are evaluated under identical environmental constraints to ensure strict fairness.

| Feature / Component | Random | AGE | CEGA | Realistic | MIME (Ours) |
|---|---|---|---|---|---|
| *Part I: Common Constraints (Fairness Protocol)* | | | | | |
| Initial Labeled Seeds | | | None (Cold Start, 0 labels) | | |
| Feedback Type | | | Hard Labels Only (Top-1 class) | | |
| Surrogate Capacity | | | GCN (Hidden Dim = 128) | | |
| Query Budget | | | Identical across all experiments | | |
| *Part II: Core Methodological Differences* | | | | | |
| Learning Type | Passive | Active Learning | Active Learning | Passive (One-shot) | **Active Learning** |
| Cold Start Strategy | Random Selection | Random Selection | Random Selection | Random Selection | **DGI / Hybrid Structural Prior** |
| Query Strategy Focus | Uniform Sampling | Unc. + Density + Centrality | Dynamic Weighted (Unc. + Div.) | Uniform Sampling | **Sequential Filtering** |
| Structure Utilization | – | PageRank | PageRank | Edge Reconstruction | **Adaptive Laplacian Regularization** |
| Post-processing | – | – | – | – | **Self-Training Fine-tuning** |

### E.3 Victim Model Baselines

Table 7 details the baseline performance of the three victim architectures (GCN, GraphSAGE, GAT) trained on the target datasets. These metrics serve as the upper bound for extraction Fidelity and explain variations in extraction Accuracy, such as the lower performance on Amazon-Computers when attacking GraphSAGE.

Table 7: Victim model accuracies (ground truth) on the global test set. Note the significantly lower and unstable performance of GraphSAGE on the AmzC dataset compared to GAT/GCN.

| Victim | Metric | CoCS | AmzC | CoraFull | Arxiv | Products |
|--------|--------|------|------|----------|-------|----------|
| GCN | Acc | $0.9382 \pm 0.0016$ | $0.8190 \pm 0.0826$ | $0.6937 \pm 0.0119$ | $0.6210 \pm 0.0039$ | $0.8727 \pm 0.0006$ |
| SAGE | Acc | $0.9424 \pm 0.0006$ | $0.6384 \pm 0.1194$ | $0.6973 \pm 0.0070$ | $0.6184 \pm 0.0059$ | $0.8779 \pm 0.0009$ |
| GAT | Acc | $0.9343 \pm 0.0016$ | $0.9014 \pm 0.0042$ | $0.7089 \pm 0.0046$ | $0.6552 \pm 0.0013$ | $0.8597 \pm 0.0048$ |

### E.4 Scalability Failure of the Realistic Attack Baseline

The time complexity of the most expensive operation in this method is approximately $O(|U| \cdot |L| \cdot D_{\text{feat}})$. When scaling from a small graph (where $|U| \approx 1,500$ and $|L| \approx 300$) to the large OGB benchmarks (where $|U| \approx 20,000$ and $|L| \approx 800$), the number of required similarity operations increases by a factor of over 100, which becomes computationally prohibitive. Since this global pairwise comparison must be repeated multiple times throughout the surrogate model training process, the total resource demand grows exponentially, rendering the method unusable under practical time constraints.

This conclusion is reinforced by the original work proposing this augmentation technique, which demonstrated success only on extremely small graph benchmarks, specifically Cora, Citeseer, and PubMed. This confirms the method was not designed for, and cannot scale to, the large-scale computational environment required for practical MLaaS platforms.

### E.5 Additional Results and Figures

This subsection contains supplementary figures and tables that provide a more comprehensive view of our experimental results, including the full performance data under varying prior sizes and the detailed ablation study outcomes.

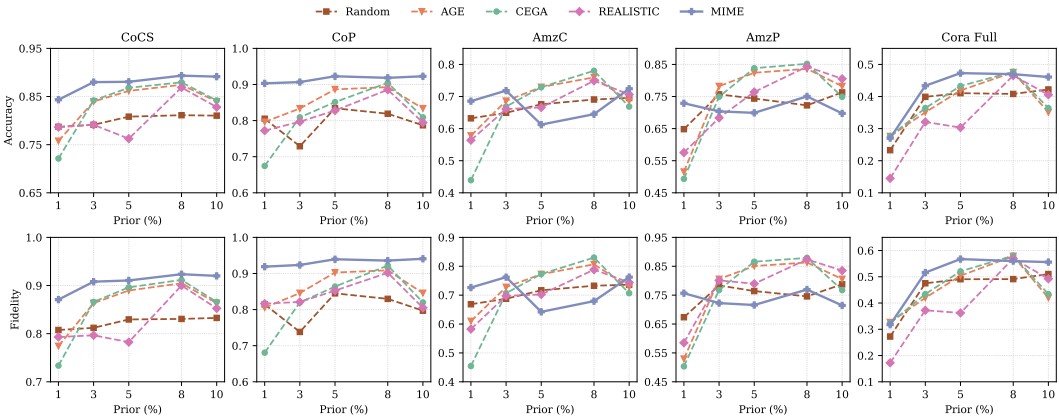

Figure 4: Complete results for accuracy and fidelity under varying prior sizes (1% to 10%) across all five benchmark datasets. These plots show the performance of MIME and all baseline methods, providing a comprehensive view of how initial information availability impacts extraction success.

Table 8: Comparison of time (s) and memory (MB) across methods and total query budgets on five datasets.

| Dataset | Method | Time (s) Query Budget | | | | Memory (MB) Query Budget | | | |
|---|---|---|---|---|---|---|---|---|---|
| | | 5C | 10C | 15C | 20C | 5C | 10C | 15C | 20C |
| CoCS | Random | 0.9 | 1.0 | 1.0 | 1.0 | 2384 | 2384 | 2384 | 2384 |
| | AGE | 2.8 | 4.5 | 6.2 | 7.9 | 2384 | 2384 | 2384 | 2384 |
| | CEGA | 1.9 | 2.8 | 3.9 | 5.3 | 2384 | 2384 | 2384 | 2384 |
| | REALISTIC | 7.5 | 7.4 | 7.2 | 7.1 | 1904 | 1904 | 1904 | 1904 |
| | MIME | 29.6 | 49.8 | 70.8 | 86.4 | 2119 | 2119 | 2119 | 2119 |
| AmazonC | Random | 0.9 | 0.8 | 0.8 | 0.8 | 2642 | 2642 | 2642 | 2642 |
| | AGE | 1.7 | 2.6 | 3.5 | 4.3 | 2642 | 2642 | 2642 | 2642 |
| | CEGA | 1.2 | 1.8 | 2.2 | 2.7 | 2642 | 2642 | 2642 | 2642 |
| | REALISTIC | 16.3 | 16.0 | 15.5 | 15.0 | 2203 | 2203 | 2203 | 2203 |
| | MIME | 17.2 | 27.4 | 40.3 | 52.6 | 1953 | 1953 | 1953 | 1953 |
| CoraFull | Random | 1.0 | 1.0 | 1.0 | 1.0 | 2642 | 2642 | 2642 | 2642 |
| | AGE | 45.0 | 89.2 | 132.5 | 175.3 | 2642 | 2642 | 2642 | 2642 |
| | CEGA | 18.2 | 45.8 | 80.5 | 123.0 | 2642 | 2642 | 2642 | 2642 |
| | REALISTIC | 7.8 | 7.1 | 6.1 | 5.0 | 2203 | 2203 | 2203 | 2203 |
| | MIME | 54.0 | 95.8 | 134.0 | 173.1 | 2316 | 2316 | 2316 | 2316 |
| Arxiv | Random | 1.1 | 1.0 | 1.0 | 1.0 | 5037 | 5037 | 5037 | 5037 |
| | AGE | 72.6 | 143.6 | 216.3 | 286.7 | 5037 | 5037 | 5037 | 5037 |
| | CEGA | 7.0 | 15.7 | 25.7 | 37.5 | 5037 | 5037 | 5037 | 5037 |
| | REALISTIC | – | – | – | – | – | – | – | – |
| | MIME | 76.5 | 128.0 | 179.4 | 227.8 | 4167 | 4167 | 4167 | 4167 |
| Products | Random | 0.8 | 0.8 | 0.9 | 0.9 | 5037 | 5037 | 5037 | 5037 |
| | AGE | 19.9 | 39.2 | 58.3 | 77.4 | 5037 | 5037 | 5037 | 5037 |
| | CEGA | 5.4 | 12.2 | 20.2 | 29.5 | 5037 | 5037 | 5037 | 5037 |
| | REALISTIC | – | – | – | – | – | – | – | – |
| | MIME | 118.5 | 201.0 | 278.4 | 357.3 | 13110 | 13110 | 13110 | 13110 |

Table 9: Detailed description of ablation study configurations. Each configuration disables or alters one component from the full MIME framework (referred to as the Baseline).

| Configuration | Component Modified | Description of Change |
|---|---|---|
| no-DGI | Unsupervised Pre-training | Skips the DGI phase; the surrogate model is randomly initialized. |
| no-Diversity | Query Selection | Removes k-center diversity logic; selects nodes on uncertainty alone. |
| no-Quota | Query Selection | Disables the per-class quota for balancing query batches. |
| no-Laplacian | Model Training | Removes the Laplacian regularizer term from the training loss. |
| no-Self-Training | Fine-tuning | Omits the final self-training step after the budget is exhausted. |
| Static Embeddings | Query Selection | Uses static embeddings from the cold start for all diversity checks. |

## F  LLM USAGE STATEMENT

During the preparation of this manuscript, we utilized a large language model (LLM) as an assistive tool. The LLM's role was primarily focused on improving the clarity and conciseness of the text. This included rephrasing sentences and paragraphs for better readability, correcting grammar, and ensuring a consistent writing style throughout the paper. Additionally, the LLM provided assistance with LaTeX formatting, helping to structure tables, figures, and other elements in accordance with the conference template.

The core research ideas, experimental design, analysis, and conclusions presented in this paper were conceived and executed entirely by the human authors. The LLM did not contribute to the research ideation. The authors have reviewed, edited, and take full responsibility for the final content and its scientific accuracy.

