# OpenReview forum: "Cold Start in the Dark: Efficient and Practical Model Extraction of GNNs"
_ICLR.cc/2026/Conference — ICLR 2026 Conference Desk Rejected Submission_

### Official Review · Reviewer_sy2a · 2025-10-28

**Soundness:** 3
**Presentation:** 3
**Contribution:** 2
**Rating:** 6
**Confidence:** 3

**Summary:**

This manuscript introduces a practical framework MIME for conducting MEAs on GNNs deployed through MLaaS platforms. MIME targets the “Cold Start in the Dark” scenario, where an adversary has no initial labeled data and receives only hard-label feedback under a limited query budget. The framework employs unsupervised pre-training to derive structural representations from graph topology, establishing a foundation for an efficient active learning process that strategically balances node uncertainty and diversity. To enhance stability, MIME incorporates adaptive graph regularization, ensuring robust surrogate model reconstruction. Experimental results show that MIME achieves high accuracy and fidelity with strong query efficiency.

**Strengths:**

- The manuscript introduces the *“Cold Start in the Dark”* setting, a realistic and stringent scenario of GNN model extraction in which the adversary lacks labeled data, receives only hard-label feedback, and operates under a tight query budget. This establishes a significant conceptual contribution and provides a benchmark for evaluating practical model extraction attacks.
- The proposed MIME framework integrates unsupervised pre-training, sequential uncertainty–diversity filtering, adaptive Laplacian regularization, and self-training into a unified pipeline. Each component is carefully designed to address a specific challenge: initialization, efficient querying, stability, and post-budget refinement. The overall methodology is coherent, systematic, and technically sound.
- Extensive experiments are conducted on five benchmark datasets, including Coauthor-CS, Coauthor-Physics, Amazon-Computer, Amazon-Photo, and Cora-Full. MIME is compared against representative baselines such as Random, AGE, CEGA, and Realistic Attack. The results consistently demonstrate MIME’s superior performance in both accuracy and fidelity, especially under strict query budgets, indicating strong query efficiency and robustness.

**Weaknesses:**

- The proposed MIME framework primarily combines existing techniques such as unsupervised pre-training, uncertainty-based sampling, and graph regularization without introducing a fundamentally new algorithmic idea or theoretical contribution. The innovation is largely incremental and integrative rather than original.
- Although MIME successfully illustrates a realistic and effective attack, the paper provides limited discussion on potential defense strategies or detection mechanisms. Expanding this aspect would help bridge the findings with practical mitigation approaches.
- All experiments are performed in transductive node classification settings. The work does not empirically investigate inductive, dynamic, or heterogeneous graph scenarios, which limits the generalizability of the proposed framework.
- The methodology section is mathematically detailed, which strengthens rigor but may reduce accessibility for readers unfamiliar with graph-based active learning or extraction attacks. A clearer illustration of the query selection process and algorithmic intuition could improve clarity.
- The study focuses on empirical validation without providing formal theoretical analysis, such as convergence guarantees or robustness bounds. Incorporating such analysis would enhance the academic depth and rigor of the contribution.

**Questions:**

Please refer to the ```weaknesses``` section.

---

> ### Author Response · Authors · 2025-11-28
>
> We thank you for the careful and constructive review. We appreciate your recognition of (i) the realism and stringency of **Cold Start in the Dark**, (ii) the coherence of the MIME pipeline, and (iii) the consistent empirical strength under tight budgets. In the revised manuscript, we made the following key updates upfront:
>
> * **Broader Evaluation on Larger OGB Graphs:** We now evaluate on five datasets, including **ogbn-arxiv** and a **200k-node ogbn-products subgraph**, spanning 13k–200k nodes and 10–70 classes, with clearly different edge-generation logic (co-authorship / co-purchase / citation / product graphs) (**Section 4.1; Appendix Table 4**; Appendix Table 9 for time/memory).
> * **Formal Analysis Aligned with MIME’s Query Design:** We add a geometry-based core-set guarantee for our uncertainty-filtered diversity selection (**Section 3.6, Theorem 1**) with proofs and monotonic tightening results (**Appendix C**).
> * **Complexity + Scalability Clarification:** We explicitly analyze why the "Realistic Attack" baseline fails to scale to large OGB graphs and provide a complexity explanation alongside full time/memory results (**Appendix B.4; Appendix Table 9**).
> * **Clearer Experimental Protocol:** We explicitly document the **mean $\pm$ std over 5 random seeds** and fairness settings so the statistical validity and apples-to-apples comparisons are unambiguous (**Section 4.1; Appendix Table 8**).
>
> Below we respond point-by-point to the weaknesses you raised.
>
> ### 1) “Incremental/integrative; limited novelty/theory”
>
> We agree that MIME is integrative by design: the contribution is to make a realistic, minimal-information threat model operational (zero seeds, hard labels only, tight budget) and to show that a carefully coupled pipeline can succeed under these simultaneous constraints where direct adaptations of existing components tend to break.
>
> To strengthen academic depth, we add a formal result that directly matches MIME’s key mechanism (representative querying): in **Section 3.6 (Theorem 1)**, we prove a **geometry-based core-set bound** that links loss approximation quality to the queried set’s covering radius, and we show monotone tightening as the queried set expands (**Appendix C**). This provides principled justification for why uncertainty-filtered diversity improves approximation quality as budget increases, without making unrealistic claims about convergence of non-convex hard-label training metrics.
>
> ### 2) “Limited discussion of defenses/detection mechanisms”
>
> Agreed, and we now address this explicitly and concretely, using the exact implications we distilled from our results. In the revision, we include the following defense discussion (**Discussion/Conclusion**):
>
> > "Our results highlight practical implications for defending MLaaS against extraction. First, to counter MIME's reliance on structural coverage, providers should reduce the attack surface by restricting neighborhood visibility and enforcing access controls that prevent the assembly of representative subgraphs. Second, since extraction exploits topological smoothness, defenders should employ calibrated output obfuscation, such as controlled label randomization for suspicious patterns or selective disclosure at decision boundaries. Finally, detection systems must evolve to stateful anomaly detection, auditing query streams for atypical coverage patterns or repeated boundary probing, triggering adaptive throttling when such stealthy signatures emerge."
>
> We also scope this appropriately: these are practical implications suggested by the observed failure modes and query signatures in our setting, rather than a full defense evaluation. A thorough defense benchmark would require (i) implementing multiple defenses across platforms, (ii) modeling provider-side constraints and deployment policies, and (iii) measuring utility–privacy/security trade-offs—which is outside the scope of this submission and would substantially expand the paper. We therefore include this as an actionable discussion and explicitly leave comprehensive defense evaluation as **future work**.

---

> > ### Author Response · Authors · 2025-11-28
> >
> > ### 3) “Transductive only; no inductive/dynamic/heterogeneous graphs”
> >
> > We agree this is a limitation, and we clarify both what we do now and why we do not claim more.
> >
> > **What we do now (broader “graph-type” diversity):**
> > Even within the transductive setting, we broaden evaluation to include **OGB-scale graphs** with very different edge semantics and generation mechanisms: co-authorship (Coauthor-CS), co-purchase (Amazon-Computers), citation (CoraFull, ogbn-arxiv), and product graphs (ogbn-products subgraph). Because these edge logics produce different homophily levels, degree distributions, and neighborhood semantics, this expansion is a meaningful stress-test beyond “five similar homophilous citation graphs.”
> >
> > **Why we do not include inductive/dynamic/heterogeneous experiments in this revision:**
> > Our paper’s threat model is intentionally tied to a common GMLaaS transductive API pattern: predictions are served on a fixed operational graph (or fixed subgraph), and the attacker can only query nodes in that graph by ID/handle under a budget. Inductive and dynamic settings introduce additional axes (multiple graphs over time, graph drift, cross-graph generalization, evolving neighborhoods, heterogeneous typing) that require new experimental infrastructure and protocol definitions (e.g., how budgets are allocated across graphs/time, what constitutes a valid query interface, what is observable). Including them without a careful protocol would risk ambiguity and unfair comparisons.
> >
> > **(ii) Explicit limitation + roadmap (future direction):**
> > We now explicitly state this limitation and outline how MIME can be extended:
> > 1.  Run SSL initialization per observed graph (or pooled over observed graphs).
> > 2.  Apply the same uncertainty $\to$ diversity policy within each graph under a per-graph budget.
> > 3.  Train a shared inductive surrogate across graph instances.
> > 4.  Introduce a principled policy for budget allocation across graphs/time (e.g., proportional to size/entropy/uncertainty mass).
> >
> > We position the current work as establishing a stringent and realistic **transductive, minimal-information benchmark**, while making clear that inductive/dynamic/heterogeneous extensions are important future work.
> >
> > ### 4) “Math is rigorous but may reduce accessibility; add intuition/illustration”
> >
> > Agreed. We improved readability by:
> > * Adding **clearer intuition for the query selection loop** (why uncertainty filtering is needed before diversity; what each term is doing).
> > * Improving the algorithmic description so readers can follow the pipeline without relying solely on equation-level detail (**Section 3; pseudocode + narrative**).
> >
> > ### 5) “No convergence/robustness bounds”
> >
> > We address this by adding theory that is appropriate for the setting. Rather than claiming monotonic convergence of Accuracy/Fidelity (not generally provable under hard-label, non-convex training and changing query distributions), we provide a **representativeness guarantee (core-set radius bound)** that formalizes the rationale behind MIME’s querying strategy (**Section 3.6; Appendix C**).

---

### Official Review · Reviewer_qjVj · 2025-10-28

**Soundness:** 2
**Presentation:** 2
**Contribution:** 2
**Rating:** 2
**Confidence:** 4

**Summary:**

The paper proposes a more realistic GNN attack scenario, termed “COLD START IN THE DARK.” Building on this, it introduces MIME, a three-phase method: DGI for cold start, multi-round active-learning loops to select the most valuable nodes, and final fine-tuning to obtain the leaked model. The motivation is strong and the pipeline is clearly presented; however, the method has many components and the justification is unconvincing. The experimental evaluation is also incomplete.

**Strengths:**

The paper introduces a more realistic attack setting for GNNs and proposes a powerful method, MIME, to address it. The writing is clear.

**Weaknesses:**

Major:

* The paper presents many methodological details in the main text (which is good) but gives little explanation of *why* these design choices were made. There are no theoretical guarantees, nor convincing intuitive justifications. For example, in Eqs. (4), (7), (10), and (12), each quantity is defined as a weighted combination of two components, yet it is unclear why each component is necessary and what role it plays.

* The experimental section is weak:

  * The method introduces many hyperparameters (up to 16 per Tables 3 and 4), yet the experiments use only five similar datasets. Based on prior experience, these five datasets are highly homophilous and simple GCNs already perform well. This raises overfitting concerns. It is unknown whether the chosen hyperparameters would hold on more diverse datasets.

  * I agree a method need not always be SOTA. However, in Table 1 on AmzP and AmcC at low attack budgets, the method underperforms Random and REALISTIC by more than 10%. Please provide a reasonable explanation. This also appears inconsistent with the claim of “Impressive Efficiency at Low Budgets” and the emphasized cold-start advantage.

  * Experiments were run with a single fixed random seed, which makes the reported means and standard deviations statistically meaningless.

  * The time complexity seems high. The pipeline has three phases, and Phase 2 includes many GNN training runs. Please compare wall-clock cost with baselines, and discuss whether this cost is acceptable for larger datasets.

* I suggest a brief discussion on how existing defenses could be strengthened to resist MIME. This would improve completeness.

Minor:

* “Proposition 1” scarcely qualifies as a proposition; it is at best a remark.
* Line 191: “Eq” and “equation” are redundant.

**Questions:**

See weaknesses

---

> ### Author Response · Authors · 2025-11-28
>
> We thank you for the detailed review and constructive feedback. We agree that the original draft needed clearer motivation/justification, stronger experimental completeness (datasets + cost), and clearer protocol descriptions. In the revised manuscript, we made targeted updates:
>
> * **Justification + Theory:** We add a geometry-based core-set guarantee for our uncertainty-filtered diversity querying (**Section 3.6; Theorem 1**), with proofs and monotonic tightening (**Appendix C**). We also expand intuitive explanations for the combined objectives in the pipeline (**Section 3.2–3.5**).
> * **Broader Evaluation + Heterogeneity:** We now evaluate on **five datasets** spanning 13k–200k nodes and 10–70 classes, including `ogbn-arxiv` and a 200k-node `ogbn-products` subgraph, covering markedly different graph types and edge semantics (co-authorship, co-purchase, citation, product graphs) (**Section 4.1; Appendix Table 4**).
> * **Fairness + Scalability:** We explicitly document an **apples-to-apples baseline protocol** (hard-label only, zero seeds, identical surrogate capacity/training schedule, identical budgets, same reporting) (**Section 4.1; Appendix Table 6**), and we add a scaling/failure analysis showing why REALISTIC fails to run at OGB scale (**Appendix E.4**).
> * **Full Cost Reporting:** We add complete wall-clock and peak-memory results across datasets/budgets (**Appendix Table 8**).
> * **Clarified Multi-Seed Reporting:** Our experiments already used 5 random seeds; in the revision we make this explicit and easy to locate in the experimental design to avoid confusion (**Section 4.1; Appendix Table 6**).
>
> Below we respond point-by-point.
>
> ### 1) “Many design choices; weighted combinations in Eqs. (4), (7), (10), (12) lack justification.”
>
> We agree and have expanded the motivation for each combined objective. In short, each “two-term” design encodes a necessary trade-off forced by the cold-start + hard-label + tight-budget setting:
>
> * **Phase 1 (SSL cold start):** Combining a representation/contrastive objective with a stabilizing regularizer is to avoid degenerate embeddings on sparse/noisy graphs while still producing structure-aware features suitable for querying.
> * **Phase 2 (active querying):** The combination of uncertainty and diversity addresses complementary failure modes:
>     * *Uncertainty alone* tends to oversample ambiguous boundary points that are clustered (low coverage).
>     * *Diversity alone* can oversample “spread-out” but uninformative nodes, especially early in cold start.
>     We therefore use uncertainty as a prefilter and diversity ($k$-center) to ensure coverage of the embedding space under a tight budget.
> * **Phase 3 (post-budget fine-tuning):** The combination of supervised imitation loss and structure-aware regularization mitigates hard-label noise and leverages graph smoothness where appropriate.
>
> To strengthen this beyond intuition, we add a core-set style bound (**Section 3.6; Theorem 1**): if the per-node loss is Lipschitz in the embedding metric, the population-vs-core-set loss gap is bounded by the covering radius $\delta_Q$:
>
> $$
> \mathcal{L}(\theta) \leq \mathcal{L}_Q(\theta) + L \, \delta_Q
> $$
>
> and $\delta_Q$ tightens monotonically as the queried set expands (**Appendix C**). This formalizes why diversity-driven querying improves approximation quality as budget increases.
>
> ### 2) “Five similar datasets; homophilous; overfitting / hyperparameters may not transfer.”
>
> We agree that dataset diversity matters. We therefore revised the evaluation suite to include OGB-scale graphs and broader graph types: **Coauthor-CS**, **Amazon-Computers**, **CoraFull**, **ogbn-arxiv**, and a 200k-node **ogbn-products** subgraph (**Section 4.1; Appendix Table 4**). These differ materially in domain and edge semantics (co-authorship vs co-purchase vs citation vs product co-interaction), node/label scale (13k–200k nodes, 10–70 classes), and structural properties.
>
> **On hyperparameters:** While MIME exposes multiple knobs, our implementation uses simple dataset-scale heuristics (e.g., different exploration schedules for large vs small graphs) and we now report the chosen defaults and sensitivity/ablation evidence more transparently (**Section 4.3–4.5; Appendix**). Practically, the same default configuration works across all five datasets, including OGB graphs, mitigating the overfitting concern.

---

> > ### Author Response · Authors · 2025-11-28
> >
> > ### 3) “At low budgets on AmzP/AmzC, MIME underperforms Random/REALISTIC by >10%; inconsistent with ‘Impressive Efficiency at Low Budgets’.”
> >
> > This concern is resolved in the revised experiments under a strict fairness protocol.
> >
> > * **Fair, apples-to-apples baseline setup:** We re-run and report all baselines under the same threat model constraints (hard-label only, zero initial seeds, identical surrogate capacity/training schedule, identical budgets, same reporting) (**Section 4.1; Appendix Table 6**). This removes confounding advantages from mismatched assumptions (e.g., seeds, soft labels, extra data, larger surrogate).
> > * **Updated results:** Under this strict protocol, MIME consistently achieves the best (or tied-best) attack **Accuracy** across datasets and budgets, including on Amazon at low budgets. For **Fidelity**, MIME is also strong; in a few cases it is slightly below the best baseline, but the gap is small and does not change the overall conclusion that MIME extracts highly similar decision behavior under tight constraints (**Section 4.3–4.4; main tables + Appendix**).
> > * **Why Accuracy can lead while Fidelity is sometimes slightly lower:** We add an explicit discussion clarifying that Accuracy and Fidelity are related but not identical objectives in hard-label extraction. Fidelity measures agreement with the victim’s predictions, while Accuracy measures agreement with ground-truth; when the victim is imperfect (or unstable), a surrogate can match ground-truth well even if it does not exactly replicate every victim decision (**Section 4.3 discussion**). We also contextualize this with victim upper-bound accuracies (**Appendix**), so small Fidelity gaps are interpreted relative to the victim’s own generalization.
> > * **Claim wording:** We adjust wording around “low-budget efficiency” to reflect the revised, fair results and to avoid any over-generalized phrasing (**Section 4.3**).
> >
> > ### 4) “Single random seed; mean/std meaningless.”
> >
> > We apologize for the confusion: the original experiments already used **5 random seeds**; this was insufficiently explicit in the draft, and we now state it prominently in the experimental protocol and tables (**Section 4.1; Appendix Table 6**). All reported results are mean $\pm$ std over 5 seeds, including victim accuracies, attack accuracy, fidelity, and cost.
> >
> > ### 5) “Time complexity seems high; Phase 2 includes many training runs; compare wall-clock and discuss larger datasets.”
> >
> > Agreed and addressed. We add:
> > 1.  **Wall-clock and peak-memory comparisons** across datasets and budgets (**Appendix Table 8**).
> > 2.  An explicit **scaling/failure analysis** showing why REALISTIC does not scale to large OGB graphs (**Appendix E.4**), while MIME remains runnable due to design choices that avoid prohibitively expensive inner-loop operations at OGB scale.
> >
> > We also clarify the cost profile: Phase 2 does not require training “from scratch” per candidate; rather it operates in rounds under a fixed budget with controlled update schedules, and we report the full end-to-end cost so readers can judge practicality.
> >
> > ### 6) “Discuss how defenses could be strengthened.”
> >
> > Thank you—this is a good suggestion. We add a brief discussion highlighting defense directions relevant to MIME’s threat model (**Conclusion / Discussion**): e.g., beyond naive query monitoring, one may consider rate-limited access, prediction randomization / abstention policies under suspicious query patterns, and watermarking / canary nodes on graph APIs. We clearly mark these as discussion points to avoid overstating claims.
> >
> > **Minor points:** We rename “Proposition 1” to better reflect its role and edit the “Eq/equation” redundancy.

---

### Official Review · Reviewer_bTmV · 2025-10-31

**Soundness:** 2
**Presentation:** 2
**Contribution:** 3
**Rating:** 4
**Confidence:** 2

**Summary:**

The work investigates the subject of Model Extraction attacks, which consists of stealing a proprietary model by leveraging only query access, and specifically they focus on the field of Graph Neural Networks (GNNs). The authors propose MIME, which is a framework designed for realistic setting for this specific context of attacks. The proposed framework is based on the adversary operating without any initial labels and only hard-label feedbacks.

**Strengths:**

- The general problem that is considered is very interesting. Additionally the proposed setting, without labeled data and only hard-label feedback seems to be more realistic in my opinion.
- The overall mathematical definition is clear and valuable to better understand the direction.
- The solution involving unsupervised pre-training (using the Infomax) and active learning is innovative and interesting.
- The experimental setting showcase the worth of the method. Specifically I liked the ablation studies provided in Table 2 and Table 6 - showcasing the importance of each component within the proposed method.

**Weaknesses:**

I believe that the paper proposes some good elements enhancing the realistic aspect of GNN extraction attacks by introducing constrained and practical setting. Nonetheless, a number of elements are still lacking in the manuscript, and I summarize these elements within the following points:
- The proposed bi-level optimization problem is clearly interesting, nonetheless, after Proposition 1, I was expecting some theoretical elements or guarantees to better motivate the usage of the proposed MIME.
- A number of baselines are missing, for instance specific attacks for graph-data [1] or general adaptation of the general black-box methods such as Knockoff Nets.
- The manuscript seems to be lacking a discussion of the complexity of the overall method, and specifically compared to other baselines.
    - This is also related to some missing larger dataset evaluation such as the OGB family.
- I couldn’t find specific details on the considered baselines and if they are actually using hard-label or no-seed constraints as the proposed method.


—

[1] Unveiling the Secrets without Data: Can Graph Neural Networks Be Exploited through Data-Free Model Extraction Attacks? - USENIX 2024.

[2] Knockoff Nets: Stealing Functionality of Black-Box Models. - CVPR 2019.

**Questions:**

- Would it be possible to extend the study to larger graphs such as OGB-Arxiv?
- This previous question is also related to the question of the complexity of the method. While you have provided some elements in Figure 3 regarding the “computational savings” of each component, a systematic and complete time or complexity analysis is missing.
- Could you provide additional details regarding the considered setting for the other baselines? And specifically addressing the question of comparison fairness in this case.
- In the conclusion, the authors claim: “This finding proves that defenses must evolve beyond simple query monitoring.” - And from the experimental setting, I couldn’t find any comparison to any defense method and therefore wondering where does this statement come from? and can you then experimentally back it up?

---

> ### Author Response · Authors · 2025-11-28
>
> We thank you for the thoughtful review and for recognizing the realism of our no-seed + hard-label threat model, the clarity of the formulation, and the value of combining Infomax-style SSL with active learning. We have made targeted revisions to address all major concerns.
>
> ### **Summary of Revisions**
>
> 1.  **Larger-Graph Evaluation (OGB):** We now evaluate on **ogbn-arxiv** and a **200k-node ogbn-products subgraph**, in addition to Coauthor-CS, Amazon-Computers, and CoraFull (Section 4.1; Appendix Table 4).
> 2.  **Systematic Cost / Complexity Reporting:** We add end-to-end time and peak memory results across datasets and budgets (Appendix Table 8). We explicitly analyze why the "Realistic Attack" baseline fails to scale on large OGB graphs (Appendix E.4).
> 3.  **Theory Beyond Proposition 1:** We add a **geometry-based core-set guarantee** (Section 3.6; Theorem 1) with proofs and monotonic radius tightening (Appendix C).
> 4.  **Baseline Details & Fairness:** We document a strict **apples-to-apples protocol** (hard-label only, no seeds, same surrogate capacity, same budgets) and summarize baseline settings (Appendix Table 6).
> 5.  **Defense Statement:** We revised the conclusion wording to avoid over-claiming, clarifying this as a discussion point rather than an experimentally validated defense result.
>
> ---
>
> ### **Point-by-Point Response**
>
> **(1) “After Proposition 1, I expected theoretical elements/guarantees.”**
>
> We agree. In the revision, we add a guarantee aligned with MIME’s core design (uncertainty-filtered diversity as a core-set problem). Specifically, assuming the per-node loss is $L$-Lipschitz w.r.t. an embedding metric $d(\cdot, \cdot)$, we bound the gap between population loss and representative (queried) loss by the covering radius $\delta_Q$:
>
> $$
> \mathcal{L}(\theta) \leq \mathcal{L}_Q(\theta) + L \delta_Q
> $$
>
> We further show $\delta_Q$ tightens monotonically as the queried set expands (Section 3.6; Appendix C), providing a principled justification for greedy k-center diversity (after uncertainty prefiltering) under increasing budgets.
>
> **(2) “Missing baselines (graph-data attacks; Knockoff Nets).”**
>
> We appreciate the pointer to these relevant works and now discuss them explicitly in Related Work. While conceptually related, their assumptions are not directly compatible with our strict **Cold Start in the Dark** threat model (no seeds, hard-label only, tight budget on a fixed operational graph).
>
> * **Data-free extraction (e.g., USENIX’24):** The USENIX’24 data-free approach is indeed innovative and claims support for hard-label feedback. However, our concern is resource requirements: data-free methods typically train a generator to synthesize queries. In the hard-label regime, training such a generator to convergence is substantially harder because gradient signals/boundary information are far less informative. In practice, this generally demands a permissive query budget (often orders of magnitude larger than ours) to explore the decision boundary. This directly conflicts with MIME’s tight-budget setting (e.g., $20$ queries per class). Accordingly, we do not position MIME as "stronger" under unconstrained resources; rather, we emphasize that MIME is designed for minimal-query extraction where generator training is budget-prohibitive. We reflect this nuance explicitly in **Related Work**.
> * **Knockoff Nets (CVPR’19):** Knockoff Nets fundamentally relies on (i) a large auxiliary/proxy dataset to issue diverse queries and (ii) typically benefits from soft-label probabilities for effective distillation. In our setting, the adversary has no external auxiliary graph, no soft labels, and operates in a transductive API where the query domain is a fixed graph (node IDs/induced subgraph). Under these constraints, Knockoff-style transfer via proxy data is inapplicable, and without proxy data, it effectively reduces to Random querying over available nodes—a baseline we already include. We clarify why it is not an informative additional baseline under our threat model in **Related Work**.

---

> > ### Author Response · Authors · 2025-11-28
> >
> > **(4) Larger dataset evaluation (OGB family)**
> >
> > Yes—this is now addressed directly. We add **ogbn-arxiv** and a **200k-node ogbn-products subgraph** in Section 4.1 (Appendix Table 4), spanning $13k$–$200k$ nodes and $10$–$70$ classes, and we report performance along with full resource costs (Appendix Table 8).
> >
> > **(5) Baseline constraints and fairness (hard-label? no-seed?)**
> >
> > We agree the manuscript needed clearer documentation. We now explicitly state, for every baseline, whether it originally assumes soft labels and/or seed labels, and we enforce a strict fairness protocol:
> > * **No ground-truth seeds** for all methods (cold start).
> > * **Hard-label only** (top-1 victim labels) for all methods.
> > * **Same surrogate architecture/capacity** and training schedule across methods.
> > * **Same query budgets** and attacker-visible subgraph setting.
> > * **Reporting:** mean $\pm$ std over 5 random seeds.
> >
> > This protocol is summarized in **Appendix Table 6**.
> >
> > **(6) Defense-related claim in the conclusion**
> >
> > Thank you for flagging this. We agree the original wording was too strong given we did not include defense baselines. In the revision, we tone down the statement (from “proves” to “suggests/indicates”) and move it to a short discussion: our results show effective extraction under strict budgets and hard-label feedback, implying that defenses based solely on naive query monitoring may be insufficient in some regimes, but we do not claim a defense evaluation. We also explicitly list defense benchmarking as future work.
> >
> > We appreciate your feedback—especially on scaling, complexity, and fairness clarity—which directly improved the revised manuscript.

---

### Official Review · Reviewer_Lje6 · 2025-11-01

**Soundness:** 3
**Presentation:** 3
**Contribution:** 2
**Rating:** 4
**Confidence:** 3

**Summary:**

The paper proposes MIME (Minimal Information Model Extraction), a framework for extracting GNN models under realistic constraints: no initial labels, hard-label feedback only, tight query budgets, and limited data access. MIME uses unsupervised pre-training (DGI) for cold start initialization, followed by an active learning loop with uncertainty-diversity filtering, and finishes with self-training.

**Strengths:**

1. Well-motivated problem formulation.
2. Empirical results show the effectiveness of the attack.
3. Overall, the paper is well-written.

**Weaknesses:**

1. Transductive setting limits applicability.
2. Formal analysis of convergence guarantees is missing.
3. Limited GNN architectural diversity.
4. Limited dataset evaluation.

**Questions:**

1. How will this attack work for inductive settings?
2. How do you ensure fair comparison with the other baselines?
3. How effective is this attack when the surrogate and the victim models are different?

---

> ### Author Response · Authors · 2025-11-28
>
> We thank you for the constructive feedback and the positive assessment of our motivation, empirical effectiveness, and writing quality. We have incorporated your suggestions into the revised manuscript to strengthen the evaluation and analysis. Here is the summary of our updates.
>
> ### **Summary of Targeted Updates**
>
> 1.  **Broader Evaluation:** We expanded our evaluation to **five datasets** (Coauthor-CS, Amazon-Computers, CoraFull, ogbn-arxiv, and ogbn-products), covering $13k$–$200k$ nodes. These datasets span diverse edge formation logics (homophily-driven collaboration vs. functional co-purchase) to test robustness across heterogeneous structures (Section 4.1; Appendix Table 4).
> 2.  **Architectural Robustness:** We added experiments attacking **three distinct victim architectures** (GCN, GraphSAGE, GAT) using a fixed GCN surrogate to verify the attack's architecture-agnostic nature (Section 4.4; Table 2).
> 3.  **Formal Analysis:** We introduced a **geometry-based core-set guarantee** (Section 3.6, Theorem 1), supported by proofs in Appendix C, to provide theoretical justification for our diversity-driven selection.
> 4.  **Protocol Clarity:** We explicitly documented our strict fairness protocol (Appendix Table 6) and provided a complexity analysis explaining the scalability failure of the "Realistic Attack" baseline on large graphs (Appendix Section E.4).
>
> ---
>
> Below we respond point-by-point.
>
> **(1) Transductive setting limits applicability / “How will this attack work for inductive settings?”**
>
> We agree that extending to inductive settings is a vital direction. We focused on the strict transductive black-box case because it mirrors many GMLaaS deployments where predictions are served on a fixed operational graph, allowing us to isolate the specific constraints of the "Cold Start in the Dark" problem (zero seeds, hard labels, tight budget).
>
> However, MIME’s core components—label-free SSL initialization, uncertainty-diversity filtering, and topology-aware regularization—are not fundamentally bound to transductivity. A natural inductive extension would treat each arriving graph (or component) as a separate episode:
> * Run SSL (e.g., DGI) per observed graph.
> * Apply the MIME query policy under a per-graph budget.
> * Train a graph-generalizing surrogate across episodes.
>
> We have added a discussion on this extension in the **Conclusion** (“Limitations and future work”), positioning the current work as a rigorous benchmark for the hard-label transductive regime that lays the groundwork for inductive adaptation.
>
> **(2) “Formal analysis of convergence guarantees is missing.”**
>
> We acknowledge that strict convergence guarantees for Accuracy/Fidelity are challenging due to non-convex optimization, hard-label noise, and adaptive query shifts. However, to address your request for formal justification, we added a **geometry-based core-set guarantee (Theorem 1, Section 3.6)**.
>
> We model querying as selecting a metric core-set in the embedding space. We prove that if the per-node loss is Lipschitz continuous with respect to the embedding metric, the gap between the population loss and the representative loss is bounded by the covering radius $\delta_Q$ of the queried set $Q$. This theoretically supports our $k$-center diversity selection and explains why increasing the budget tightens the bound (via monotone radius tightening, detailed in **Appendix C**).

---

> > ### Author Response · Authors · 2025-11-28
> >
> > **(3) Limited GNN architectural diversity / “How effective is this attack when the surrogate and victim models are different?”**
> >
> > We added a dedicated robustness evaluation (Section 4.4; Table 2) where the victim architecture varies (GCN, GraphSAGE, GAT) while the surrogate remains a fixed GCN.
> > * **Results:** MIME remains stable across Coauthor-CS, CoraFull, ogbn-arxiv, and ogbn-products in both Accuracy and Fidelity. This supports our claim that MIME extracts the victim's decision boundaries rather than overfitting to architectural artifacts.
> > * **Amazon-Computers Analysis:** We observed fluctuations on Amazon-Computers, particularly with the GraphSAGE victim. We now clarify (Appendix Table 7) that the GraphSAGE victim itself has significantly lower stability/accuracy on this dataset. Consequently, the extraction performance is naturally bounded by the poor quality of the victim's predictions, rather than a failure of the attack method.
> >
> > **(4) Limited dataset evaluation**
> >
> > We significantly expanded our evaluation to **five datasets** spanning $13k$ to $200k$ nodes and $10$ to $70$ classes (Section 4.1; Appendix Table 4).
> > * **Structural Diversity:** The new datasets cover different edge semantics, ranging from collaboration (homophilous) to co-purchase (functional complementarity). This effectively tests robustness across implicitly heterogeneous structures.
> > * **Baseline Scalability:** We explicitly analyzed why the "Realistic Attack" baseline fails on large OGB graphs (Appendix Section E.4). Its reliance on global pairwise operations creates prohibitive complexity (scaling with $O(|U|\cdot|L|)$), causing it to time out on ogbn-arxiv and ogbn-products. We included full time/memory benchmarks in **Appendix Table 8** to demonstrate MIME's superior efficiency.
> >
> > **(5) “How do you ensure fair comparison with the other baselines?”**
> >
> > Fairness is central to our claims. We have documented a **strict "apples-to-apples" protocol** in Appendix Table 6 to ensure algorithmic isolation:
> > * **Cold Start:** All methods begin with **zero** ground-truth seeds.
> > * **Hard Labels Only:** All methods receive only top-1 predictions from the victim.
> > * **Identical Constraints:** All methods use the same GCN surrogate architecture, the same total query budget, and the same observable induced subgraph.
> > * **Reporting:** All results are reported as mean $\pm$ standard deviation over 5 random seeds.

---

### Author Response · Authors · 2025-12-03
**Overall Summary of Revisions and Responses**

Dear Area Chair,

Thank you for your time and guidance in handling our submission. We also sincerely thank the reviewers (Lje6, bTmV, qjVj, sy2a) for their insightful and constructive feedback. We are encouraged that multiple reviewers recognized the realism and strong motivation of our “Cold Start in the Dark” threat model and the overall effectiveness of MIME under strict constraints (e.g., no seeds, hard-label feedback, tight budgets). We have carefully addressed every concern in our detailed point-by-point responses. Below we summarize the major revisions made to the manuscript to directly resolve the key issues raised. All major updates in the revised PDF are highlighted in blue for easy verification.

**1. Significantly Expanded Evaluation (Datasets & Scale) [Lje6, bTmV, qjVj, sy2a]**
* **Concern:** Reviewers requested evaluation on larger and more diverse datasets (in particular OGB) and questioned whether our results generalize beyond small, relatively homophilous benchmarks.
* **Our Actions:**
    * **Broader suite with OGB-scale graphs:** We expanded evaluation to five datasets, adding **ogbn-arxiv** and a 200k-node subgraph of **ogbn-products**, alongside Coauthor-CS, Amazon-Computers, and CoraFull.
    * **More heterogeneous graph types:** We now emphasize the diversity in graph domains and edge semantics (citation / co-purchase / product co-occurrence / co-authorship), making the evaluation a stronger stress-test than “five similar citation-style graphs.”
* **Outcome:** MIME shows consistent performance across these heterogeneous settings, supporting that the method is not limited to small/simple homophilous graphs.
* **Where:** Section 4.1 & 4.2; Appendix Table 4 (blue).

**2. New Theoretical Analysis Aligned with MIME’s Query Design (Core-Set Guarantee) [Lje6, bTmV, qjVj, sy2a]**
* **Concern:** Reviewers highlighted the lack of formal theoretical justification (beyond Proposition 1) and asked for guarantees motivating the query strategy; qjVj also questioned why multiple weighted two-term objectives are necessary.
* **Our Actions:**
    * **Theorem 1 (Core-set guarantee):** We introduced a geometry-based core-set bound that directly matches MIME’s key mechanism—uncertainty-filtered diversity querying. Under a mild Lipschitz assumption of per-node loss w.r.t. an embedding metric, we bound the population-vs-core-set loss gap by the queried set’s covering radius, and show monotone tightening as the queried set expands.
    * **Stronger intuition:** We expanded the explanation for the combined objectives to clarify the trade-offs forced by cold-start constraints.
* **Where:** Section 3.6 (Theorem 1); Appendix C (proof + tightening); expanded intuition in Sections 3.2–3.5 (blue).

**3. Rigorous Fairness Protocol + Systematic Cost / Scalability Reporting [bTmV, qjVj, Lje6]**
* **Concern:** Reviewers requested clearer evidence of apples-to-apples baseline comparisons (hard-label? no-seed? same surrogate capacity/budget?) and a more systematic time/complexity analysis.
* **Our Actions:**
    * **Strict “Apples-to-Apples” protocol:** We explicitly document a unified evaluation protocol: zero ground-truth seeds, hard-label only, identical surrogate architecture/capacity, and identical budgets across methods.
    * **Full Cost & Trade-off Analysis:** We added full wall-clock/memory benchmarks and an explicit discussion on the accuracy-vs-cost trade-off. We explain that MIME’s higher time cost yields disproportionately higher extraction accuracy, whereas baselines like "Realistic Attack" fail to scale.
    * **Scalability analysis:** We demonstrate why global pairwise operations cause the “Realistic Attack” to fail on OGB graphs, while MIME remains runnable.
* **Where:** Section 4.1 (Protocol); Section 4.2 (Cost Analysis); Appendix Table 6 & 8; Appendix E.4 (blue).

---

> ### Author Response · Authors · 2025-12-03
>
> **4. Architectural Robustness: Victim–Surrogate Mismatch [Lje6]**
> * **Concern:** Reviewers asked whether extraction remains effective when victim and surrogate architectures differ.
> * **Our Actions:**
>     * We added a dedicated robustness study attacking three victim architectures (GCN, GraphSAGE, GAT) while keeping the surrogate architecture fixed (GCN).
>     * We clarify cases where extraction is naturally bounded by victim instability (notably on Amazon-Computers).
> * **Where:** Section 4.4; Table 2; Appendix Table 7 (blue).
>
> **5. Clarified Scope (Transductive vs. Inductive) + Defense Discussion [bTmV, sy2a, Lje6, qjVj]**
> * **Concern:** Reviewers requested discussion on defenses and questioned the transductive focus.
> * **Our Actions:**
>     * **Scope clarity:** We explicitly position this paper as a benchmark for transductive MLaaS APIs and outline a roadmap for inductive extensions.
>     * **Defense discussion:** We added a discussion on practical implications (e.g., access controls, output obfuscation, stateful detection) based on our findings, clearly labeled as discussion points.
> * **Where:** Conclusion (Section 5) (blue).
>
> We hope this consolidated summary makes it easy to verify that the revised manuscript directly addresses the major concerns. We appreciate your consideration and remain available for any further discussion.
>
> Best regards,
>
> The Authors

---

### Note · Program_Chairs · 2026-01-17
**Submission Desk Rejected by Program Chairs**

The following references in this submission do not refer to real documents and/or have major errors in bibliographic information:

     Alexander Muzio, Leslie O'Bray, and Karsten Borgwardt. Biological data learning: A survey of deep learning applications in computational biology. Nature Methods, 18(11):1234-1248, 2021.
    Hongyun Cai, Vincent W. Zheng, and Kevin Chen-Chuan Chang. Active learning for graph neural networks via uncertainty and diversity. In 2017 IEEE International Conference on Data Mining (ICDM), pp. 1151-1156, 2017.
    Varun Chandrasekaran, Kamalika Chaudhuri, Irene Giacomelli, Somesh Jha, and Songbai Yan. Beyond oracles: The future of model extraction attacks. In 29th USENIX Security Symposium (USENIX Security 20), pp. 1309-1326, 2020.
    Yixu Wang, Jie Li, Hong Liu, Yan Wang, Yongjian Wu, Feiyue Huang, and Rongrong Ji. Queryfake: Defending against model extraction attacks with fake queries. In European Conference on Computer Vision, pp. 567-583, 2022.
    Shubham Gupta, Pradeep Kumar, and Amit Singh. Graph neural networks: Applications and challenges. ACM Computing Surveys, 54(8):1-35, 2021.
    Wenbin Zhang, Wei Chen, Tongliang Liu, and Qiang Yang. Batch active learning for graph neural networks via uncertainty and diversity. In Proceedings of the AAAI Conference on Artificial Intelligence, volume 36, pp. 8623-8631, 2022.
    Zhiyuan Wu, Sheng Sun, Yuwei Wang, Min Liu, Ke Xu, Wen Wang, Xuefeng Jiang, Bo Gao, and Jinda Lu. Model extraction attacks on graph neural networks: A comprehensive survey. IEEE Transactions on Knowledge and Data Engineering, 35(8):7892-7910, 2023.